# Cytomimetic calcification in chemically self-regulated prototissues

Rui Sun[1,2], Zhuping Yin[1], Molly M. Stevens [2,3], Mei Li [1] ✉ & Stephen Mann [1,4] ✉

The fabrication of cytomimetic materials capable of orchestrated and adaptive functions remains a significant challenge in bottom-up synthetic biology. Inspired by the cell/matrix integration of living bone, here we covalently tether distributed single populations of alkaline phosphatase-containing inorganic protocells (colloidosomes) onto a crosslinked organic network to establish viscoelastic tissue-like micro-composites. The prototissues are endogenously calcified with site-specific mineralization modalities involving selective intra-protocellular calcification, matrix-specific extra-protocellular calcification or gradient calcification. To mirror the interplay between osteoblasts and osteoclasts, we prepare integrated prototissues comprising a binary population of enzymatically active colloidosomes capable of endogenous calcification and decalcification and utilize chemical inputs to induce structural remodelling. Overall, our methodology opens a route to the chemically self-regulated calcification of homogeneous and gradient tissue-like mineral-matrix composites, advances the development of bottom-up synthetic biology in chemical materials research, and could provide potential opportunities in bioinspired tissue engineering, hydrogel technologies and bone biomimetics.

Given the recent development of a wide range of synthetic cell models (protocells) based on liposomes[1–3], polymersomes[4,5], inorganic colloidosomes[6,7], organoclay/DNA capsules[8], proteinosomes[9], and membraneless[10–13] and membranized[14–17] coacervate droplets, methods for the chemical construction of artificial tissue-like constructs (prototissues) comprising interconnected or distributed aggregates of protocells are currently being explored[18–20]. Prototissues have been constructed in aqueous media via surface contact-dependent interactions involving electrostatic forces[21], covalent bonding[19,22] or DNA complementarity,[23] or through external physical force patterning using 3D printing[18], acoustic[24] and magnetic fields[25], or optical tweezers[26], and collective behaviours such as signal processing[23,27], chemo-mechanical deformation[18,19] and enzyme-mediated metabolism[24] reported. As a step towards mimicking living cell/extra-cellular matrix interactions, protocell populations have been physically immobilized within viscoelastic polysaccharide hydrogels to produce soft tissue-like modules that exhibit signal-induced chemo-mechanical movement[28], chemical signal processing[27,29], and parallel transmission of electrochemical inputs[30].

Recent advances in the fabrication of bone-like composites have involved ion-doped nano-hydroxyapatite and chitosan scaffolds[31], biomineralized bioglass scaffolds[32], hierarchical biomineral structures[33] and the nanoscale control of mineralisation[34]. From a cytomimetic perspective, previous studies have demonstrated that model protocells can be used as microscale scaffolds for studies on bioinspired mineralization. For example, coacervate micro-droplets have been used as molecularly crowded environments for the deposition of silica[35] and calcium carbonate/phosphate[36], and lipid[37] and polymer[38] vesicles employed as micro-reactors for calcium carbonate precipitation. Cell-free expression of biomineralization

[1]Centre for Protolife Research and Centre for Organized Matter Chemistry, School of Chemistry, University of Bristol, Bristol, UK. [2]Department of Medical Biochemistry and Biophysics, Karolinska Institute, Stockholm, Sweden. [3]Department of Physiology, Anatomy and Genetics, Department of Engineering Science, Kavli Institute for Nanoscience Discovery, University of Oxford, Oxford, UK. [4]Max Planck-Bristol Centre for Minimal Biology, School of Chemistry, University of Bristol, Bristol, UK. ✉e-mail: mei.li@bristol.ac.uk; s.mann@bristol.ac.uk

enzymes has been undertaken to initiate biomineralization within polymeric artificial cells[39] and calcium phosphate colloidosomes used to induce bone formation in vitro and in vivo[40]. Although these studies demonstrate the potential of individual protocells to spatially confine inorganic nucleation and growth to microscale dimensions, the construction of macroscale mineralized prototissues by the chemical self-transformation of integrated protocell/matrix communities remains a significant challenge.

Prototissues capable of mechanical and chemical communication have been developed by matrix-assisted assembly of protocells[41]. Here we develop strategies that mirror the cell/matrix integration of living composites such as bone, in which biological construction and remodelling depend critically on the coordinated interplay between specialized cells (osteoblasts, osteoclasts) and their extracellular matrix (collagen)[42,43]. Osteoblasts secrete bone matrix and small extracellular vesicles that are enzymatically active in generating inorganic phosphate ions. The influx and accumulation of phosphate ions and $Ca^{2+}$ through membrane transporters induce the nucleation and subsequent growth of calcium phosphate inside the extracellular vesicles[44]. Confinement within the vesicles together with the presence of the extracellular matrix operate synergistically to produce a mineralized tissue comprising an integrated network of living cells, organic matrix and inorganic nanocrystals. Osteoclasts, in contrast, promote mineral/matrix dissolution and degradation that are necessary for bone resorption and homoeostasis.

To represent bone tissue processing within a synthetic chemistry context, we covalently link a homogeneously distributed population of alkaline phosphatase (ALP)-containing inorganic protocells (colloidosomes) to a crosslinked calcium alginate hydrogel network to establish viscoelastic prototissues that can be endogenously transformed in the presence of calcium glycerophosphate (CaGP) into calcified microcomposites. We present three different modalities of calcium phosphate mineralization involving simultaneous or differential protocell/matrix calcification, including selective intra-protocellular calcification, matrix-specific extra-protocellular calcification and gradient calcification. To mimic the interplay between osteoblasts and osteoclasts intrinsic to bone remodelling, a binary population of colloidosomes capable of implementing a cycle of endogenous ALP-mediated calcification[39,45] and esterase-induced decalcification is homogeneously distributed within the prototissues and activated in the presence of CaGP (phosphate production) or ethyl acetate (H+ production), respectively. Taken together, our strategy provides a step to cytomimetic materials based on the self-regulated calcification of soft tissue-like microstructures and offers a general approach to the fabrication of integrated protocell/matrix micro-composites with potential applications in hydrogel technologies, bioinspired tissue engineering and bone biomimetics.

## Results

### Prototissue construction and calcification

Protocell/matrix-integrated prototissues in the form of self-standing circular disks (diameter ($d$) × thickness ($t$), 4.5 × 1.6/1.8/2.0 mm), rectangular strips (12 × 2 × 2 mm) or thin circular sheets ($d × t = 4.5$ mm × 260 μm) were constructed using a multistep fabrication process (Fig. 1a). Enzyme-active colloidosomes (mean diameter, 60 μm) with a methacrylate (MA)-functionalized semi-permeable silica membrane and internal silica network were prepared by emulsifying an ALP-containing buffer solution (typically 2 mg ml⁻¹ ALP in 0.1 M Tris, pH=8.5) with partially hydrophobic silica nanoparticles in anhydrous dodecane, followed by crosslinking the Pickering emulsions with tetramethyl orthosilicate (TMOS) and 3-(trimethoxysilyl)propyl methacrylate (TPM) and transferring the microcapsules into water (Supplementary Fig. 1). Characterization studies confirmed that ALP remained encapsulated after transfer and that MA was grafted to the silica membrane (Supplementary Fig. 2). Although the MA-modified

membrane was permeable to free ALP in the external environment, leakage of encapsulated ALP from the colloidosomes was minimal (Supplementary Fig. 3) due to irreversible binding of the enzyme to the silica membrane and internal silica network. MA-functionalized colloidosomes were covalently integrated at high number density into a MA-modified alginate hydrogel (Alg-MA, 75 kDa) by photo-assisted protocell/matrix tethering and matrix crosslinking. Ionic crosslinks between alginate chains were then introduced by addition of calcium ions ($Ca^{2+}$) to produce an artificial tissue-like structure (Fig. 1b). Typically, the colloidosome volume fraction was ca. $61 \pm 12\%$ equivalent to ca. $181 \pm 22.5$ protocells/mm². The immobilized MA-colloidosomes were structurally intact and showed no significant change in diameter after integration into the hydrogel (Fig. 1c). Rheological measurements revealed an increase in storage modulus compared with analogous hydrogels fabricated with non-modified colloidosomes (Fig. 1d and Supplementary Fig. 4), indicating that chemical integration of the protocells into the Alg-MA matrix increased the stiffness of the viscoelastic micro-composites. In general, the above methodology was highly reproducible and could be adapted to fabricate various tissue-like constructs (Supplementary Fig. 5), as well as readily extended to the construction of multifunctional prototissues composed of binary protocell populations arranged in spatially homogeneous (Fig. 1e) or gradient microstructures (Fig. 1f).

Having established that viscoelastic tissue-like micro-composites with high levels of colloidosome/matrix integration could be reproducibly constructed, we sought to transform the covalently linked networks into calcified prototissues using a cytomimetic strategy loosely analogous to bone mineralization. As ca. 99% of the total ALP concentration remained associated with the MA-colloidosomes after encapsulation, we exploited the embedded protocells as spatially distributed reaction hotspots for the endogenous generation of phosphate ions and subsequent initiation of calcium phosphate nucleation within the prototissues (Fig. 2a). Prototissue disks were incubated in an aqueous solution of CaGP at room temperature ($25 \pm 0.5$ °C) to provide a reservoir of free $Ca^{2+}$ ions and initiate ALP-mediated hydrolysis of GP to generate phosphate ions within the tissue-like constructs (Supplementary Fig. 6). Consequently, the translucent prototissue became progressively opaque over 24 h (Fig. 2b and Supplementary Fig. 7). Confocal laser scanning microscopy (CLSM) images and scanning electron microscopy/energy dispersive X-ray (SEM/EDX) elemental (Ca/P/Si) mapping of the calcified samples showed that calcium phosphate particles were present within the colloidosomes and throughout the Alg-MA matrix (Fig. 2c, d and Supplementary Fig. 8). The Ca/P ratio (1.84) was higher than reported ratios for amorphous calcium phosphate (1.0-1.5) and hydroxyapatite (1.67), possibly because of excess $Ca^{2+}$ ions trapped in the Alg-MA matrix. Cryo-SEM was utilized to visualize the sub-microscale structure and texture of the calcified prototissues and revealed networks of calcium phosphate particles inside and outside the colloidosomes that were different in mean particle size and network density. After 24 h, mineralization within the colloidosomes – that is, in closer proximity to the site of ALP-mediated phosphate production – resulted in calcified silica networks that were relatively dense and consisted of aggregated calcium phosphate particles, approximately 200 nm in mean size (Fig. 2e, f). In contrast, efflux of excess phosphate into the surrounding Alg-MA matrix produced smaller calcium phosphate aggregates (mean size, ca. 145 nm) that were closely associated with the continuous polysaccharide network (Fig. 2e, f).

Cryo-SEM images indicated that calcium phosphate nucleation occurred mainly within the colloidosome interior and adjacent matrix within 0.5 h of CaGP addition, followed by nucleation throughout the extra-protocelluar matrix within 3 h (Fig. 2g and Supplementary Fig. 9). The mineral content increased progressively to a threshold value within 12 h of CaGP addition (Fig. 2h) and reached a plateau at 24 h.

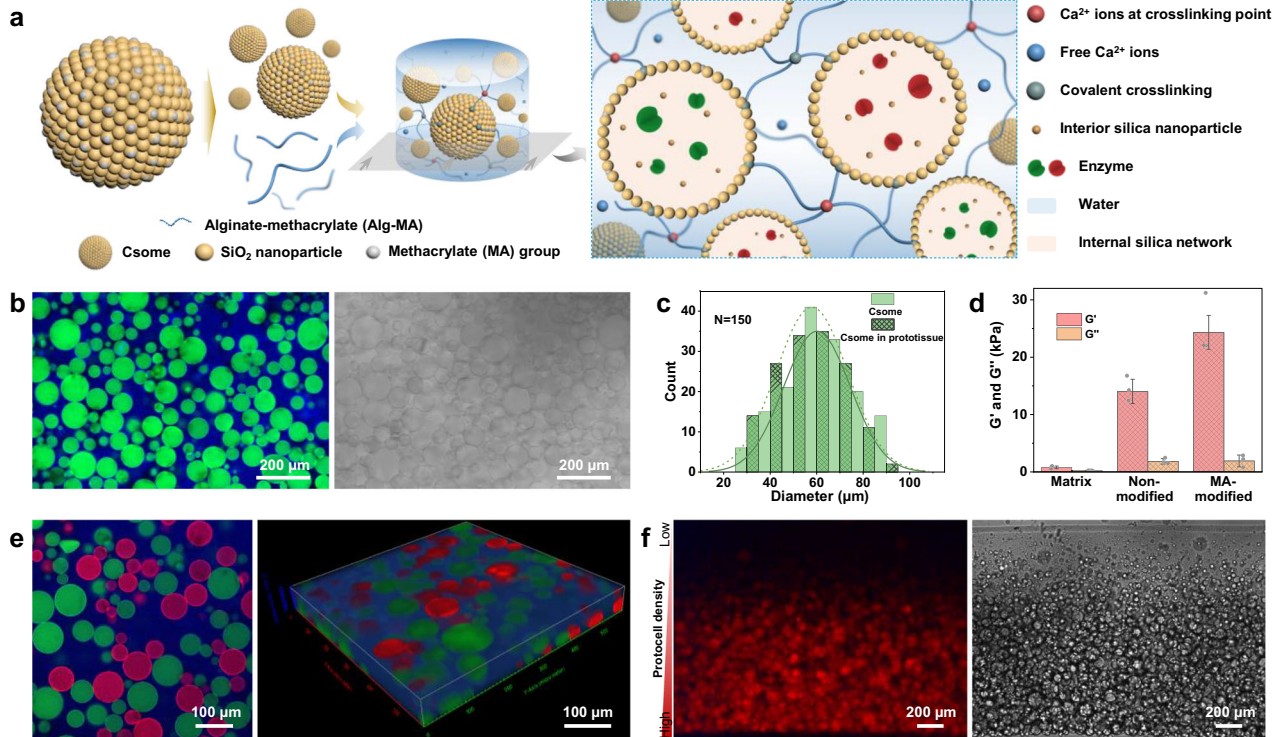

**Fig. 1 | Construction of protocell/matrix-integrated model prototissues.**
**a** Chemical integration of colloidosomes into a crosslinked alginate hydrogel. Left panel: enzyme-containing colloidosomes (Csome) comprising crosslinked silica nanoparticles (yellow) functionalized with methacrylate (MA) (light grey) are mixed with MA-modified alginate (Alg-MA) (blue lines) in the presence of $Ca^{2+}$ ions and exposed to UV light to produce a prototissue disk with spatially localized enzyme activity. Right panel: colloidosomes (ring of yellow dots) with encapsulated enzymes (red/green symbols) and residual silica nanoparticles (dark-yellow dots) are covalently linked to the extra-protocellular Alg-MA matrix (blue lines) via MA groups on the protocell membrane. The Alg-MA hydrogel is covalently (MA, dark teal) and ionically ($Ca^{2+}$, red) crosslinked and contains free $Ca^{2+}$ ions (blue). **b** CLSM fluorescence (left) and bright field (right) images of a protocell/matrix-integrated prototissue showing enzyme-containing colloidosomes (green fluorescence, fluorescein isothiocyanate (FITC)-labelled ALP) immobilized within a cross-linked Alg-MA matrix (blue fluorescence, Dylight 405-labelled Alg). No significant leakage of ALP or penetration of Alg-MA into the colloidosomes is observed. Scale bars,

200 μm. **c** Plots of colloidosome (Csome) diameters before and after matrix immobilization showing retention of protocell size. **d** Rheology measurements showing increased storage modulus ($G'$) for prototissues comprising a covalently integrated MA-colloidosome/Alg-MA matrix (MA-modified) compared with a non-integrated colloidosome/Alg-MA matrix (non-modified) or crosslinked Alg-MA without colloidosomes (matrix). Data are presented as mean values ± s.d. ($n = 3$ samples). **e** Fluorescence CLSM image (left) of a prototissue containing a binary population of colloidosomes with encapsulated FITC-labelled ALP (green channel) or RITC-labelled bovine serum albumin (BSA) (red channel). The reconstructed 3-D fluorescence CLSM image (right) shows a homogeneous spatial distribution of the different colloidosomes within the Alg-MA matrix. Scale bars, 100 μm. **f** Corresponding fluorescence microscopy (left) and bright field (right) images of a prototissue showing a gradient distribution of RITC-labelled ALP-encapsulated colloidosomes within the Alg-MA matrix. The red colour scale indicates the protocell density from low density (light red) to high density (dark red). Scale bars, 200 μm. Source data are provided as a Source Data file.

Thermogravimetric analysis of samples prepared for 12 or 24 h gave an organic mass fraction of ca. 27 or 26 wt.% and calcium phosphate content of ca. 22 or 24 wt.%, respectively, at an ALP concentration of 2 mg ml⁻¹ (Supplementary Table 1). Increasing the concentration of ALP trapped within the colloidosomes significantly increased the mineral content of the calcified prototissues to give a maximum mineral fraction of ca. 60 wt.% at an ALP concentration of 10 mg ml⁻¹ (Supplementary Fig. 10, 11, and Supplementary Table 1). The corresponding maximum mineral volume fraction was estimated to be around 10 vol.% (10.21 ± 0.86 vol.%) based on data obtained from confocal microscopy, cryo-SEM, and mass and density measurements. The calcium phosphate yield per volume of the prototissue was estimated as 0.04953 ± 0.005 g/cm³. Other experiments indicated that the mineral content was only increased slightly by refreshing the CaGP solution every 24 h for up to 3 days.

Structural investigations based on time-dependent Raman spectra and powder X-ray diffraction (PXRD) profiles indicated that the as-prepared calcified prototissues consisted of amorphous calcium phosphate (ACP) (Supplementary Fig. 12-13). The ACP content

remained essentially constant after immersion in CaGP solution for up to 28 d (Supplementary Fig. 12). In contrast, immersing the calcified prototissues in pure water for 28 d resulted in the phase transformation of ACP to poorly crystalline hydroxyapatite (HAP) (Supplementary Fig. 12-14). Unconfined compression measurements on the calcified prototissues showed stress-strain curves consisting of an initial non-linear region that exhibited a slow response in stress to increasing strain, suggestive of a viscoelastic response, followed by an elastic linear region up to a stress of ca. 160 kPa, and plastic deformation at higher stress values (Supplementary Fig. 15). The Young's moduli ($E$) were determined from the linear regions of the stress-strain curves and showed increased levels of stiffness after calcification. In particular, the $E$ value for the non-mineralized prototissue (0.2 ± 0.04 MPa) was increased to 1.13 ± 0.09 MPa and 2.20 ± 0.33 MPa when mineralized with ACP (24 h, 24 wt.%) or HAP (35 d, 24 wt.%), respectively ([ALP] = 2 mg ml⁻¹) (Supplementary Fig. 16). In contrast, protocell/matrix-integrated composites prepared with mineral levels as high as 60 wt.% ([ALP] = 10 mg ml⁻¹) were brittle due to the relatively high inorganic content.

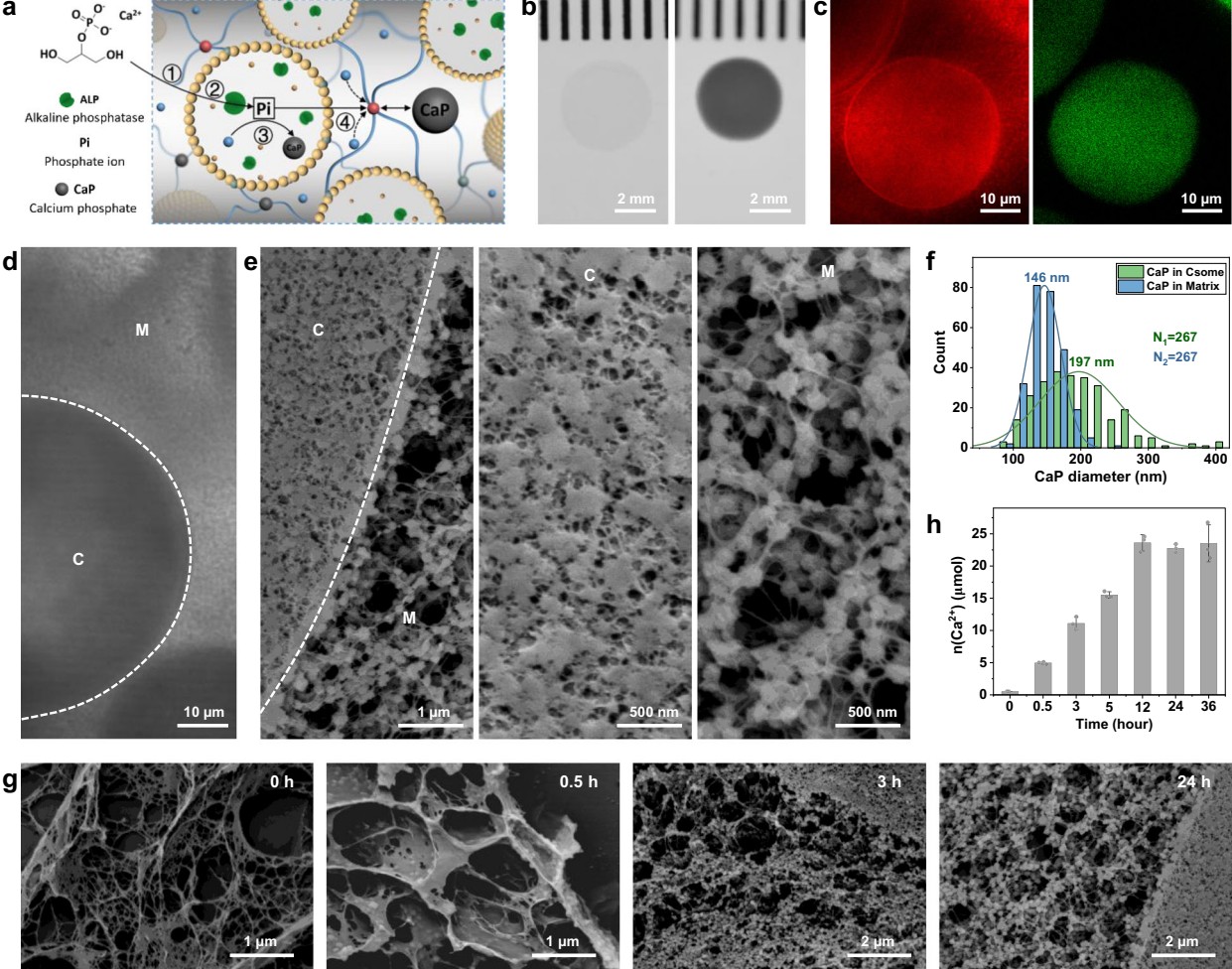

**Fig. 2 | Calcification in protocell/matrix-integrated prototissues. a** Scheme: *1*, Diffusion of CaGP into crosslinked Alg-MA and uptake of CaGP into a tethered ALP (green symbol)-containing MA-colloidosome (ring of yellow dots); *2*, ALP-mediated GP hydrolysis and localized production of inorganic phosphate (Pi); *3*, Interaction of Pi with free $Ca^{2+}$ ions (blue dots) initiates intra-protocell deposition of calcium phosphate (CaP) (dark grey spheres) under supersaturation conditions; *4*, Efflux of excess Pi promotes CaP nucleation in the extra-protocellular matrix. Other symbols as described in Fig. 1a. **b** Photographs of a pristine translucent prototissue disk (left) and after immersion in CaGP solution for 24 h to produce a calcified prototissue (right). Scale bar 2 mm. **c** Corresponding fluorescence CLSM images showing CaP within the colloidosomes and crosslinked Alg-MA matrix (left; ARS-stained CaP, red channel) and retention of FITC-labelled ALP within the protocells (right; green channel). Images were recorded 24 h after addition of CaGP. Scale bars, 10 μm. **d** Bright field image of a thin section of a calcified prototissue after 24 h showing

differences in contrast (mineralization) at the colloidosome/matrix (C/M) interface (white dashed lines). Scale bar, 10 μm. **e** Left; Cryo-SEM image of a freeze-fractured calcified prototissue after 24 h showing a calcified silica network in C and calcified alginate nanofilaments in M. Middle/right; magnified images of the C and M regions, respectively. Scale bars; 1 μm, 500 nm, 500 nm. **f** Size distribution plots for CaP particles deposited within the colloidosomes (CaP in Csome; mean = 197 μm) and surrounding Alg-MA matrix (CaP in matrix, mean = 146 μm) of a calcified prototissue. **g** Freeze-fractured cryo-SEM backscattered electron images showing time-dependent nucleation and growth of CaP in the extra-protocellular matrix of a calcifying prototissue. Scale bars from left to right; 1, 1, 2 and 2 μm. **h** Plot of amount of $Ca^{2+}$ determined from acid-extracted prototissues against calcification time; $Ca^{2+}$ measurements are a proxy for the amount of CaP deposited. Data are presented as mean values ± s.d. (*n* = 3 samples). Source data are provided as a Source Data file.

## Site-specific calcification in protocell/matrix-integrated prototissues

Given that simultaneous mineralization in both the discontinuous (protocell) and continuous (matrix) phases of the prototissues could be achieved by the propagation of ALP-mediated phosphate reaction-diffusion gradients arising from covalently tethered colloidosomes, we implemented chemical strategies to generate prototissues exhibiting either intra- or extra-protocellular calcification. Specifically, we constructed prototissues with site-specific environments capable of promoting or inhibiting calcium phosphate deposition within localized domains of the protocell/matrix-integrated prototissues.

Deposition of calcium phosphate solely within the interior of the colloidosomes was accomplished indirectly by inhibiting ACP nucleation in the surrounding hydrogel matrix. To achieve this, we replaced

the Alg-MA hydrogel with a poly(ethylene glycol, PEG) matrix produced by photo-assisted crosslinking of PEG dimethacrylate (PEGDM, $M_w$ = 750; monomer units, n = 13). PEGDM was used as initial test experiments indicated that the incorporation of increasing amounts of PEG in a series of integrated colloidosome/Alg-MA prototissues progressively curtailed the level of calcification in the hybrid hydrogels (Supplementary Fig. 17–19). By completely replacing Alg-MA with PEGDM and initiating endogenous ALP-mediated phosphate production in the presence of CaGP for 24 h, covalently integrated protocell/matrix prototissues with spatially discrete colloidosome-based discontinuous mineralization domains and a mineral-free crosslinked matrix were constructed (Fig. 3a–c and Supplementary Fig. 20-21). High magnification CLSM and bright field microscopy images (Fig. 3d), and corresponding fluorescence intensity and grey scale line profiles

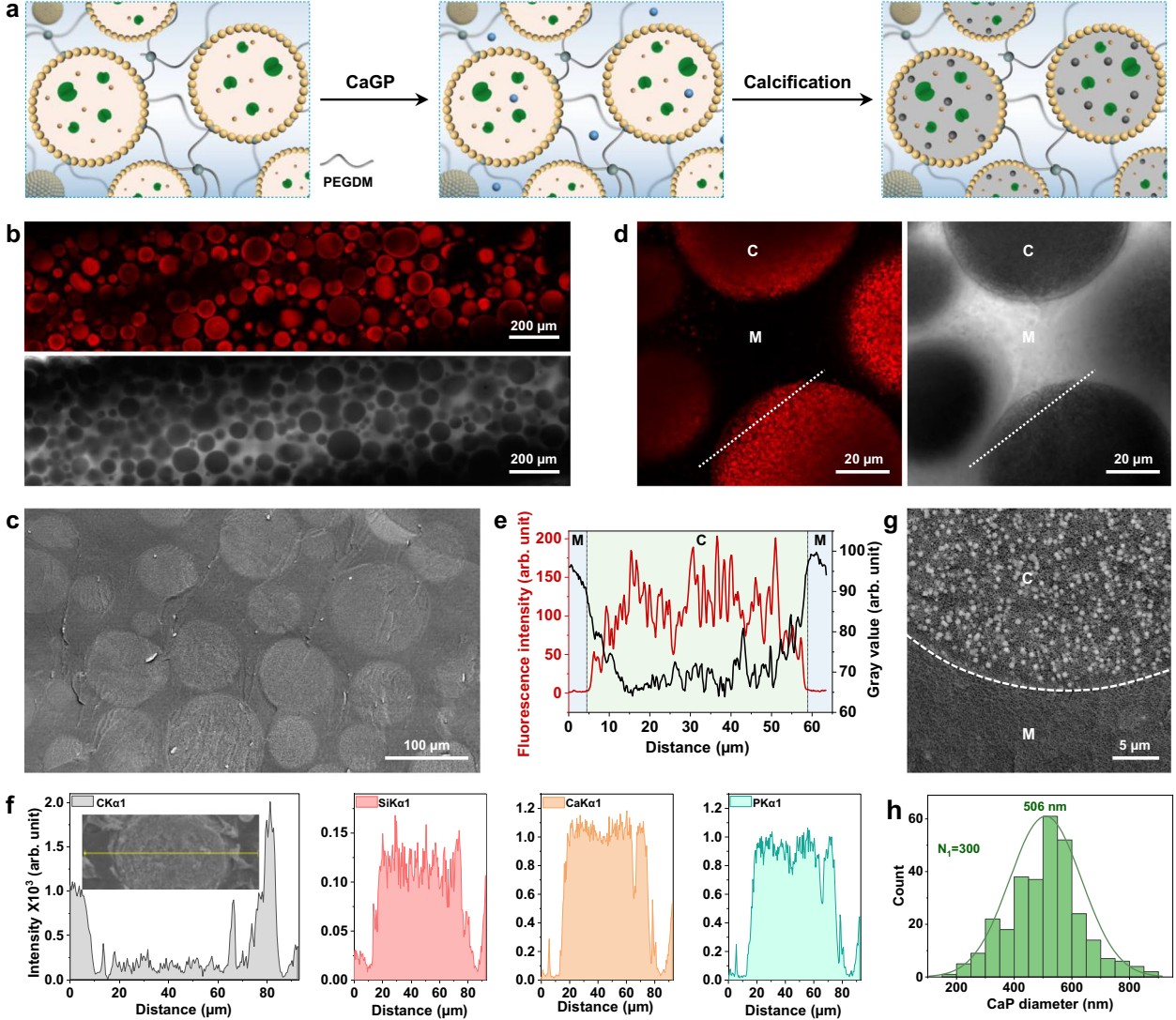

**Fig. 3 | Selective intra-protocellular calcification. a** Scheme for intra-protocellular calcification in protocell/matrix-integrated prototissues via matrix-mediated inhibition. Left panel: MA-colloidosomes (rings of yellow dots) are covalently tethered into a crosslinked (dark teal dots) PEGDM (grey lines) extra-protocellular matrix by photo-polymerization. Middle panel: addition of CaGP induces endogenous ALP (green symbols)-mediated inorganic phosphate (Pi) production. Right panel: Calcification (grey shading) for 24 h occurs only within the colloidosomes in the presence of free Ca²⁺ions (blue dots). PEG-mediated inhibition of ACP nucleation results in negligible mineralization in the PEGDM matrix. **b** Corresponding fluorescence (top) and bright field (bottom) CLSM images of a calcified prototissue disk exhibiting discontinuous mineralization domains associated specifically with the population of spatially distributed colloidosomes (ARS-stained ACP, red fluorescence (top image); opaque regions (bottom image). The continuous crosslinked PEGDM matrix displays minimal red fluorescence (top) and remains translucent (bottom) due to an absence of calcification. Scale bars, 200 μm. (**c**) Cryo-SEM backscattered electron image of a freeze-sectioned calcified

prototissue showing mineralized colloidosomes (light grey domains) embedded in a continuous mineral-free PEGDM matrix. Scale bar, 100 μm. **d** Corresponding high magnification fluorescence (left) and bright field (right) CLSM images showing site-specific calcification within colloidosomes (C, ARS-stained ACP, red fluorescence) and mineral-free matrix (M). Scale bars, 20 μm. **e** Corresponding line profiles (white dashed line in d) showing red fluorescence intensity (ARS-stained ACP) and grey scale traces recorded across the colloidosome/matrix interface. Blue shaded areas: matrix (M); green shaded area: colloidosome (C). **f** SEM-EDX spectroscopy line profiles recorded across an individual calcified colloidosome (insert) showing site-specific distributions of carbon, silicon, calcium and phosphorus. **g** Cryo-SEM backscattered electron image of the interface between an individual calcified colloidosome (C) and surrounding PEGDM matrix (M, dark region) showing ACP nanoparticles within the model protocell. Scale bar, 5 μm. **h** Size distribution plot for intra-colloidosome ACP particles. All samples were calcified for 24 h. Source data are provided as a Source Data file.

(Fig. 3e), along with SEM-EDX line scan analyses (Fig. 3f), confirmed that calcium phosphate deposition was specifically localized within the MA-colloidosomes and not in the surrounding PEGDM matrix. The Ca/P elemental ratio was 1.6, consistent with the lower levels of Ca²⁺ in the sample. PXRD and Raman spectroscopy data indicated that the intra-protocellular mineral phase was ACP (Supplementary Fig. 22), and high magnification cryo-SEM images showed that the silica-containing colloidosomes were packed with spherical ACP particles with a mean diameter of 506 nm (Fig. 3g, h and Supplementary Fig. 21a). The

particle size was almost three-times larger than observed for the ACP aggregates deposited under comparable conditions in the MA-colloidosome/Alg-MA prototissues (see Fig. 2f). The increased diameter of the colloidosome-containing ACP aggregates was attributed to inhibition of calcification in the surrounding crosslinked PEGDM matrix such that mineral growth within the protocells was significantly more competitive compared with in the external environment. The stress-strain curves for the MA-colloidosome/PEGDM-integrated prototissues showed evidence for viscoelasticity along with increased levels

of stiffness when mineralized specifically inside the protocells with ACP (24 h) or HAP (35 d in water) ($E = 3.81 \pm 0.52$ and $4.18 \pm 0.32$ MPa, respectively) (Supplementary Fig. 23, 24) compared with homogeneously calcified MA-colloidosome/Alg-MA prototissues prepared under similar conditions ($E = 1.13 \pm 0.09$ (24 h) and $2.20 \pm 0.33$ MPa (35 d)), respectively (see Supplementary Fig. 16). This was attributed to the increased stiffness of the non-mineralized MA-colloidosome/PEGDM hydrogel matrix ($E = 1.33 \pm 0.33$ MPa, unmineralized; MA-colloidosome/Alg-MA matrix, $E = 0.2 \pm 0.04$ MPa), and increased density of the MA-colloidosome/PEGDM organic matrix (Supplementary Table 1 and Supplementary Fig. 21b).

Having established a methodology for selective intra-protocellular mineralization via matrix-mediated inhibition, we developed an alternative strategy to construct prototissues comprising a continuous calcified extra-protocellular Alg-MA matrix and non-calcified ALP-containing colloidosomes. Specifically, a carboxylate-rich polymer (polyacrylic acid, PAA, 100 kDa) was included in the Alg-MA hydrogel to promote calcification within the matrix while PEG (100 kDa) was encapsulated within the ALP-containing MA-colloidosomes to inhibit intra-protocell nucleation and growth of ACP (Fig. 4a). Initial test experiments indicated that increasing the PAA content of an Alg-MA hydrogel matrix progressively promoted site-specific calcification in the extra-protocellular space after addition of CaGP (Supplementary Fig. 25) and that encapsulation of PEG (100 kDa) increased the level of matrix mineralization in Alg-MA-based prototissues prepared without PAA by inhibiting calcification at the site of endogenous phosphate production (Supplementary Fig. 26). Given these observations, we combined these approaches under optimum conditions (Alg-MA : PAA = 30: 70 weight ratio; $[PEG]_{IN} = 40$ mg ml$^{-1}$) to construct covalently integrated protocell/matrix-integrated prototissue disks with high levels of mineralization preferentially located in the continuous extra-protocellular medium (Fig. 4b,c and Supplementary Fig. 27). Cryo-SEM images indicated that the immobilized colloidosomes remained structurally intact and exhibited localized surface folding (Fig. 4d), indicating that the protocells remained relatively soft and elastic compared with the adjacent calcified matrix. Cross-sectional images showed a dense network of calcium phosphate particles with a mean size of 160 nm that extended throughout the Alg-MA/PAA matrix (Fig. 4e, f), and which were identified as ACP by PXRD and Raman spectroscopy (Supplementary Fig. 28). SEM-EDX elemental mapping gave a Ca/P ratio of 2.0 and indicated that the calcium phosphate phase was preferentially localized in the extra-protocellular matrix (Fig. 4g) with the Ca intensity in the Alg-MA/PAA matrix almost two-times higher than in the MA-colloidosomes (Fig. 4h). Unconfined compression measurements confirmed that the mineral-matrix prototissues showed viscoelastic behaviour with stress-strain curves similar to micro-composites comprising selective intra-protocellular mineralization (Supplementary Fig. 29). Storing the calcified prototissues in water for 35 days transformed the ACP particles into poorly ordered HAP (Supplementary Fig. 28) with a concomitant increase in the Young's moduli from $1.84 \pm 0.28$ to $2.76 \pm 0.25$ MPa (Supplementary Fig. 29).

Taken together, the above results indicate that viscoelastic protocell/matrix-integrated prototissues with different calcified microscale organizations can be prepared by fine-tuning the chemical properties of the colloidosome interior and extra-protocellular matrix. As a result, the Young's modulus of the pristine prototissue (0.2 MPa), which was comparable to the soft dermis tissue of skin (ca. 0.1–0.2 MPa)[46] increased to values of $1.13 \pm 0.09$, $3.81 \pm 0.52$ and $1.84 \pm 0.28$ MPa in as-prepared calcified prototissues with homogeneous protocell/matrix mineral distributions, selective intra-protocellular deposition or selective matrix mineralization, respectively (Supplementary Fig. 30). These values increased respectively to $2.20 \pm 0.33$, $4.18 \pm 0.32$ and $2.76 \pm 0.25$ MPa after in situ transformation of ACP to HAP, commensurate with the typical stiffness determined for biological cartilage tissue[46,47]. In general, the mechanical properties of

the different calcified prototissues were associated with correlations between the Young's modulus and organic : inorganic ratio, indicating a synergistic effect between matrix density and mineral content.

## Matrix-specific calcification in gradient prototissues

Inspired by the possibility of controlling the matrix-specific localization of calcification within the prototissues, we sought to develop this system towards the fabrication of protocellular-based composites capable of endogenously generating spatial gradients of calcification within a tissue-like environment. Gradient prototissues in the form of rectangular strips were prepared by partial sedimentation of a population of ALP/PEG-containing MA-colloidosomes in an aqueous mixture of Alg-MA and PAA, followed by photo-assisted matrix covalent crosslinking and protocell/matrix tethering (Fig. 5a). Protocell/matrix-integrated prototissue microstructures with a continuous number density gradient or discontinuous segregation of colloidosomes were prepared depending on the sedimentation time used (Fig. 5b–d). In both cases, calcified replicas could be readily fabricated by the addition of CaGP. Cryo-SEM and SEM-EDX element mapping analysis confirmed the occurrence of matrix-specific calcification and showed a progressive change (Fig. 5e and Supplementary Fig. 31) or a distinct discontinuity (Supplementary Figs. 32,33) in the degree of mineralization in the gradient or segregated prototissues, respectively.

The systematic change in matrix content across the height of unmineralized or calcified gradient prototissue rectangular strips was utilized to trigger a chemo-mechanical response in the spatially heterogeneous micro-composites (Fig. 5f). Typically, the prototissue strips, which became opaque 10 min after addition of aqueous 0.05 M CaCl$_2$ or CaGP due to increased ionic crosslinking of the matrix, deformed symmetrically away from the long axis towards the matrix-enriched side of the gradient over a period of 120 min (Fig. 5g and Supplementary Movie 1 and Supplementary Fig. 34). The degree of curvature increased as the gradient/segregated microstructures became more established (Supplementary Fig. 35, 36). Bright field images of selected areas of the bent strips confirmed that the deformation was directed away from the colloidosome-enriched side of the gradient (Fig. 5h). We attributed the chemo-mechanical bending to differential shrinkage (Supplementary Fig. 37) and changing mechanical stiffness associated with the different levels of Ca$^{2+}$-induced ionic crosslinking along the protocell/matrix-integrated gradient (Supplementary Fig. 38, 39).[48] Interestingly, simultaneously inducing the onset of differential matrix shrinkage and endogenous phosphatase (mineralization) activity within the embedded ALP/PEG-containing MA-colloidosomes by addition of CaGP reduced the rate and extent of shape deformation compared with unmineralized gradient prototissues exposed to CaCl$_2$ (Fig. 5i), suggesting that the increased stiffness due to incipient calcification impeded the chemo-mechanical dynamics. Similarly, exogenously induced removal of Ca$^{2+}$ ions from the deformed calcified or Ca-crosslinked (unmineralized) gradient prototissues resulted in different rates of reversibility (Fig. 5j and Supplementary Fig. 40). Specifically, linear rectangular strips were re-established within 23 min by exposure of the mineral-free Ca-crosslinked curved prototissues to 0.05 M ethylenediaminetetraacetic acid (EDTA), while corresponding experiments with bent calcified gradient prototissues showed a similar behaviour but with a slower relaxation time (ca. 72 min) (Fig. 5j, Supplementary Fig. 40 and Supplementary Movie 2).

## Calcification remodelling in multi-protocellular prototissues

As a step towards mimicking the dynamic interplay of osteoblasts and osteoclasts in bone remodelling,[43] we constructed synthetic prototissues with a community of chemically integrated protocells capable of either endogenous enzyme-mediated calcification or decalcification (Fig. 6a). Specifically, in place of a single population of MA-colloidosomes, a 1 : 1 binary population of MA-colloidosomes containing either

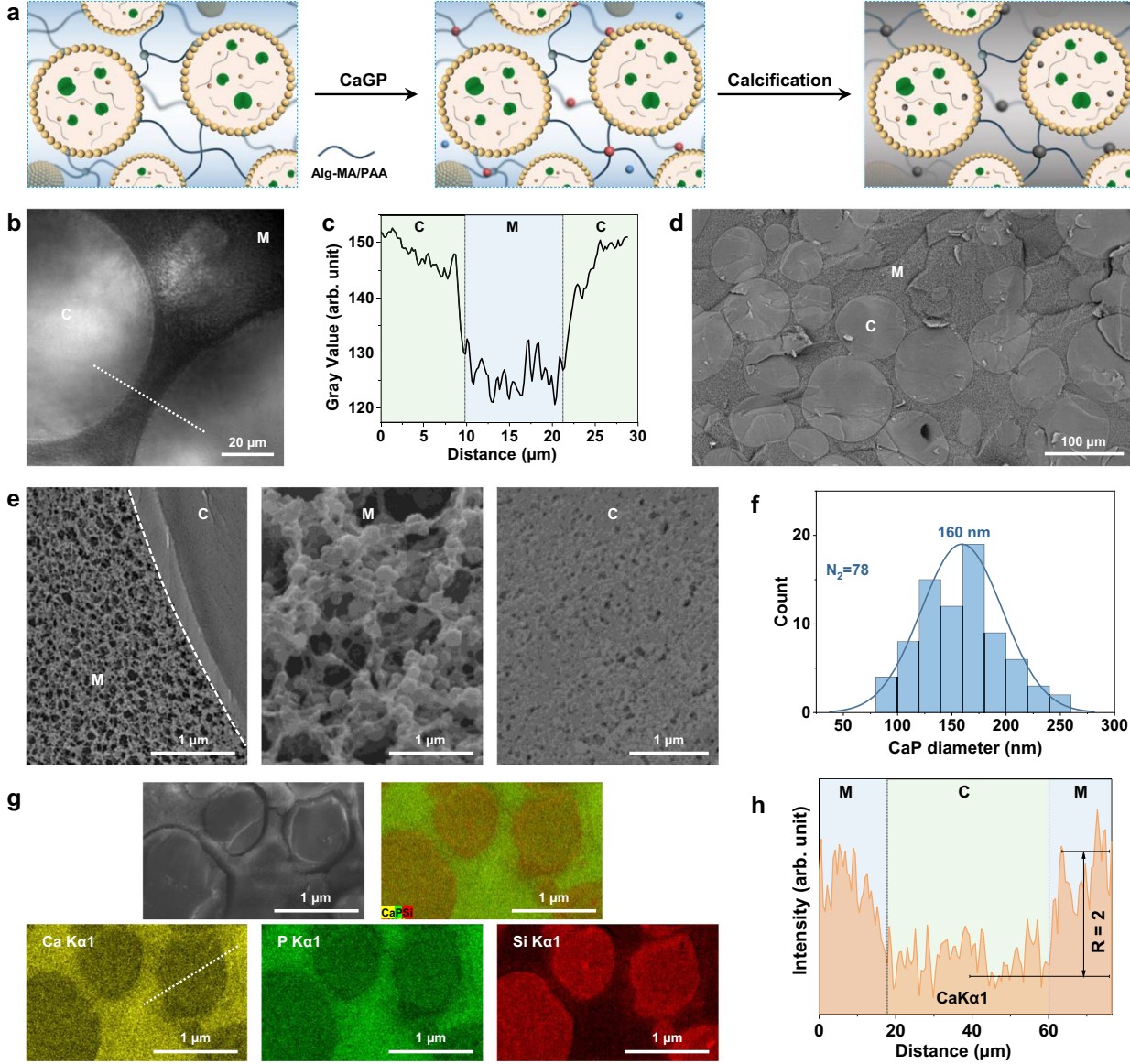

**Fig. 4 | Selective extra-protocellular matrix calcification. a** Scheme for extra-protocellular calcification in protocell/matrix-integrated prototissues by spatially separated synergistic enzyme processing. Left panel: PEG (thin grey lines)-containing MA-colloidosomes (rings of yellow dots) are covalently tethered into a crosslinked (dark teal dots) Alg-MA/PAA (30: 70 wt.%; dark-blue lines) matrix. Middle panel: addition of CaGP induces endogenous ALP (green symbols)-mediated inorganic phosphate (Pi) production. Right panel: Calcification (grey shading) occurs preferentially within the Alg-MA/PAA matrix due to increased sequestration of free $Ca^{2+}$ (blue dots) by PAA (bound $Ca^{2+}$ (red dots)) and PEG-mediated inhibition of ACP nucleation within the colloidosomes. **b** Bright field image showing relatively high and low light transmittance in the colloidosome (C) and matrix (M) domains of a calcified prototissue, respectively, indicative of selective extra-protocellular mineralization; scale bar, 20 μm. **c** Corresponding grey value line trace (white dashed line in (**b**)). Blue shaded area: matrix (M); green shaded areas: colloidosome (C). Cryo-SEM backscattered electron images recorded at low (**d**) and high (**e**) magnification of a calcified prototissue exhibiting preferential matrix mineralization. The colloidosomes (C) exhibit a smooth texture at low magnification compared with the roughness of the matrix (M) (**d**). High magnification images recorded across the C/M interface (**e**, left) show a heavily calcified matrix (middle) and silica/PEG sub-structure inside the colloidosomes (right). Scale bars, 100 μm (**d**); 1 μm (**e**). **f** Size distribution plot for extra-protocellular ACP particles; mean = 160 nm. **g** SEM-EDX element mapping showing site-specific distribution of calcium/ phosphorus and silicon in the extra-protocellular matrix and within the colloidosome interior, respectively. Scale bar, 1 μm. **h** Corresponding line profiles of calcium recorded across the colloidosome/matrix interface (white dashed line in (**g**)). All samples were calcified for 24 h. Source data are provided as a Source Data file.

ALP (2 mg mL⁻¹) or esterase (40 mg mL⁻¹) was homogeneously integrated into an Alg-MA prototissue disk (Fig. 6b), and CaGP was added to induce ALP-mediated calcification (Fig. 6c, d). Cryo-SEM images and SEM-EDX elemental mapping recorded after 24 h revealed that ACP particles were distributed in the crosslinked matrix and within the interior of both the ALP-loaded and esterase-containing colloidosomes, with the former being more extensively calcified (Fig. 6e and Supplementary Fig. 41). To induce decalcification within the multi-protocellular prototissues, ethyl acetate (EA) was diffused into the

mineralized disks at room temperature (25 ± 0.5 °C) to switch on the local production of protons specifically within the esterase-containing colloidosomes. Consequently, the pH of the external solution decreased from around 8 to a value of 6 over 32 h (Fig. 6f), during which the prototissue disk progressively changed in appearance from opaque to translucent, becoming comparable with the pristine (non-mineralized) prototissue (Supplementary Fig. 42). CLSM images revealed that the mean gray values determined from the different colloidosomes in the pristine and decalcified prototissues were

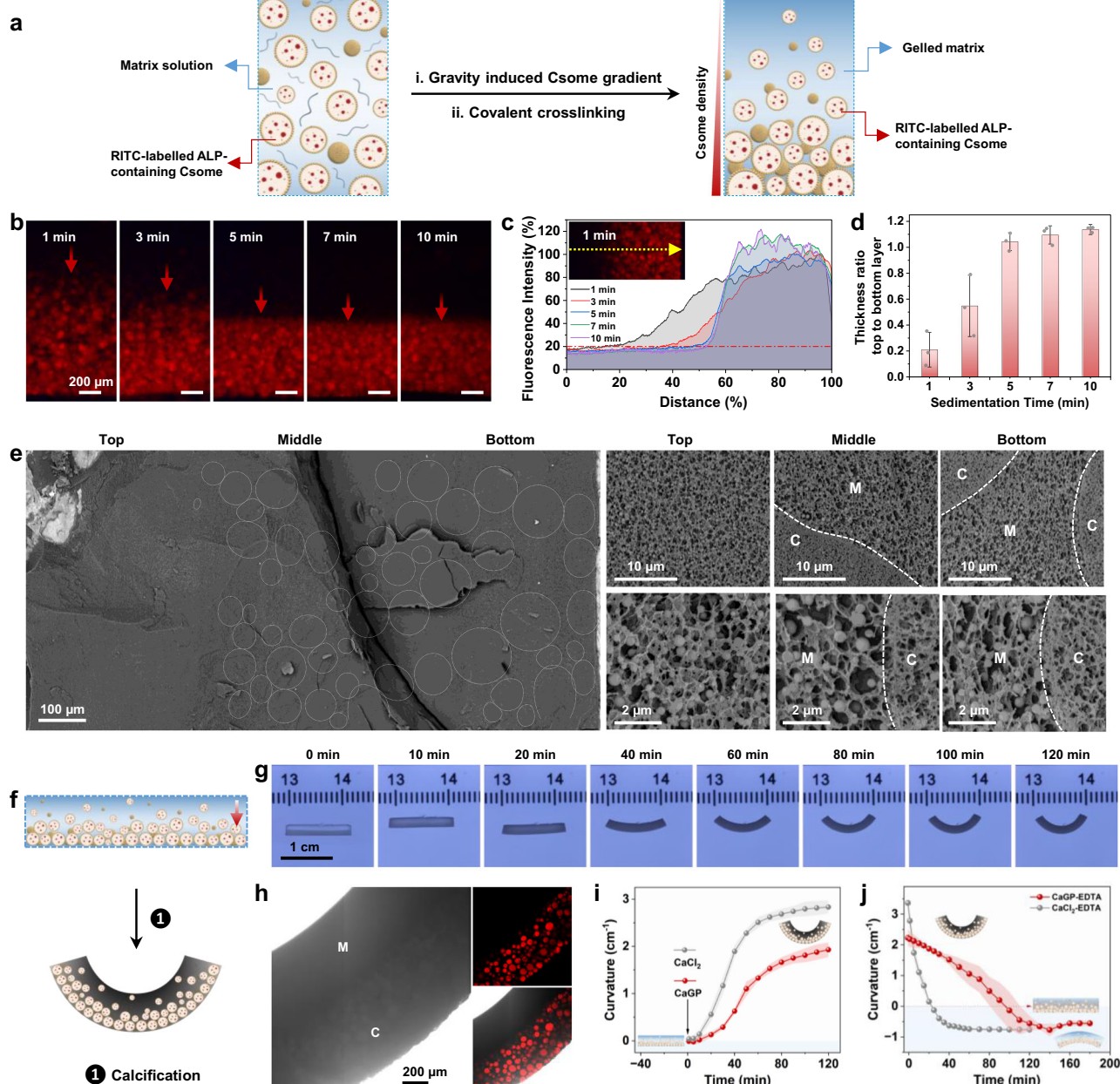

**Fig. 5 | Matrix-specific calcification in chemo-mechanically active gradient prototissues. a** Construction of a gradient structured prototissue: (i) ALP/PEG-containing MA-colloidosomes (Csome) and matrix precursors (Alg-MA and PAA) are partially sedimented. (ii) The resulting protocell gradient is immobilized in the matrix by photo-assisted covalent crosslinking. **b** Fluorescence microscopy images of different gradient prototissues prepared from colloidosome/Alg-MA/PAA suspensions sedimented for 1, 3, 5, 7 or 10 min and then covalently crosslinked. Red arrows show the direction of sedimentation. Scale bars, 200 μm. (**c**) Plot of fluorescence intensity distribution across a gradient prototissue recorded along the direction of sedimentation (yellow dashed arrow in insert). The upper/lower layer boundary is 20% of maximum fluorescence intensity. **d** Plot of ratio of upper layer thickness (matrix-enriched) to lower layer thickness (colloidosome-enriched) after different sedimentation times. Data are presented as mean values ± s.d. (*n* = 3 samples). **e** Cryo-SEM images. Left: low magnification image showing top/middle/bottom regions of a calcified gradient prototissue; MA-colloidosome (white circles), scale bar, 100 μm. Right: medium (upper row) and high (lower row) magnification images of selected regions; scale bars, 10 and 2

μm, respectively. M, matrix; C, colloidosome. **f** Calcification and chemo-mechanical deformation in a gradient prototissue. Top: rectangular strip with protocell gradient (red arrow). Immersion in CaGP (1) promotes Ca²⁺-mediated bending and endogenous calcification (bottom). **g** Time-series of photographs of a deforming calcifying gradient prototissue strip after addition of CaGP. Large graduation marks, 1 cm. **h** Left, bright field image of a gradient prototissue (M, matrix-enriched; C, colloidosome-enriched region). Darker regions show higher levels of calcification and matrix crosslinking. Top right, corresponding CLSM image (red fluorescence, RITC-labelled ALP-colloidosomes). Bottom right, merged bright field/fluorescence image. Scale bar, 200 μm. (**i**) Plots of increasing curvature against time from 40 min before (−40 to 0, *x* axis) to 120 min after addition of CaGP to a gradient prototissue strip undergoing deformation and calcification (red), or after immersion in CaCl₂ (deformation only, grey). Data are presented as mean values ± s.d. (*n* = 3 samples). **j** Plots of decreasing curvature against time after addition of EDTA to deformed prototissue strips in the presence of CaGP (red) or CaCl₂ (grey). Data are presented as mean values ± s.d. (*n* = 3 samples). Source data are provided as a Source Data file.

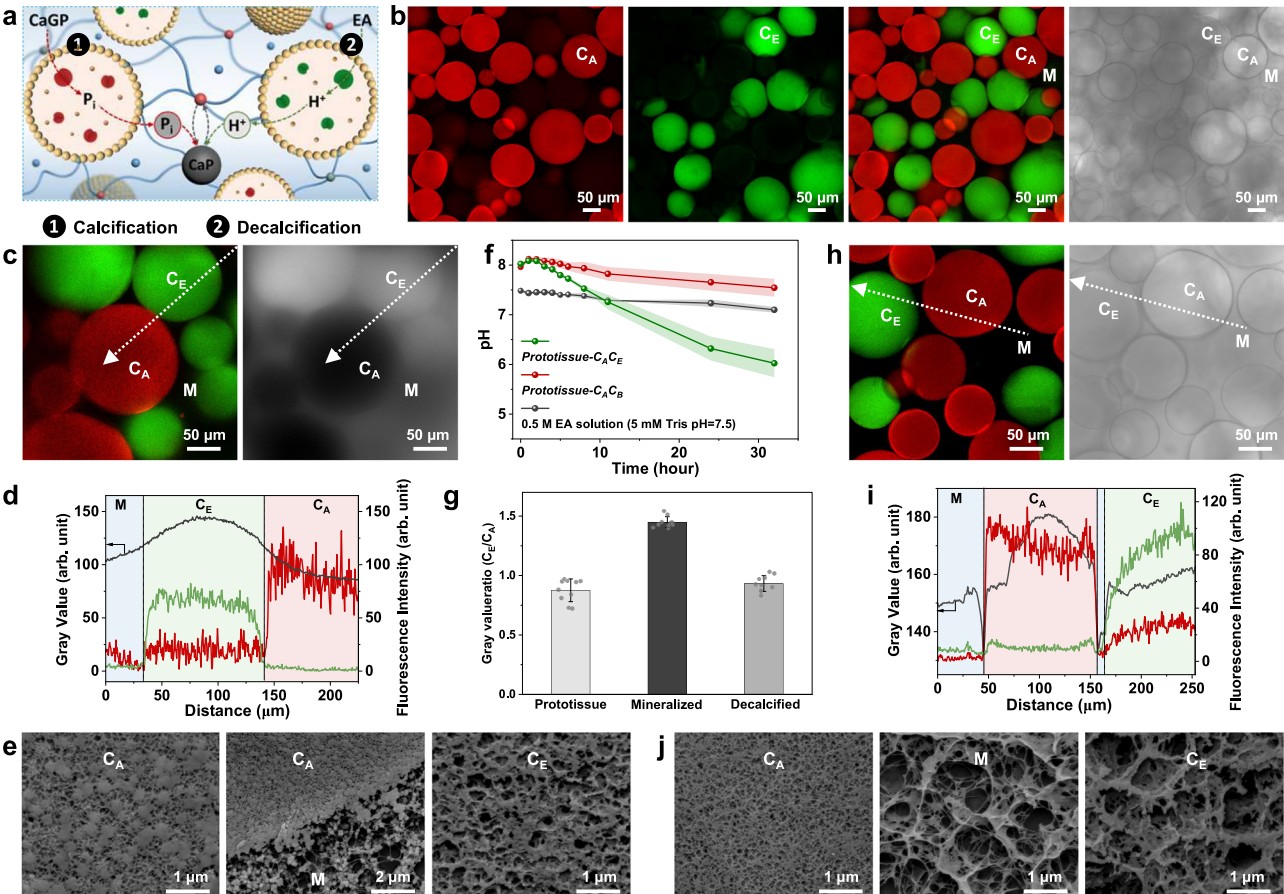

**Fig. 6 | Calcification remodelling in multi-protocellular prototissues.**
**a** Calcification/decalcification sequence in an Alg-MA/PEGDM matrix containing MA-colloidosomes with entrapped ALP (red) or esterase (green). 1, Calcification; uptake of CaGP, ALP-mediated production/efflux of phosphate, and mineralization (CaP). 2, Decalcification; uptake of EA, esterase-mediated acidification, and demineralization. Free $Ca^{2+}$ (blue dots); bound $Ca^{2+}$ (red dots). **b** Fluorescence/bright field (BF) CLSM images of an as-prepared prototissue containing RITC (red)-ALP-active colloidosomes ($C_A$) and FITC (green)-esterase-active colloidosomes ($C_E$). Black regions, non-fluorescent matrix (M). Left to right; red, green, red/green channels, BF image. Scale bar, 50 μm. **c** Left, fluorescence CLSM image of a calcified prototissue with $C_A$ (red), $C_E$ (green) and matrix (black) domains. Right, corresponding BF image showing selectively mineralized $C_A$ (dark regions). Scale bar, 50 μm. **d** Fluorescence intensity/grey value line profiles (white line in (**c**)). ALP and esterase remain spatially separated; $C_E$ exhibits low levels of mineralization (highest grey value). Blue shaded area: matrix (M); green shaded area: $C_E$ protocell; red shaded area: $C_A$ protocell. Background levels of red fluorescence are observed in $C_E$ due to autofluorescence. (**e**) Cryo-SEM images of a freeze-fractured calcified prototissue recorded within the interior of $C_A$ (left), across the $C_A$/M interface and

within $C_E$ (right). Mineralization in the matrix is curtailed due to the inhibitory effect of PEGDM. Scale bars from left to right; 2, 2 and 1 μm. **f** Time-dependent plots of pH after addition of EA to calcified prototissues prepared from ALP- and esterase-containing colloidosomes (*prototissue-$C_AC_E$*, green line), or ALP- and BSA- colloidosomes (*prototissue-$C_AC_B$*, red line), control). Reference sample, EA in Tris buffer (dark grey line). Data are presented as mean values ± s.d. (*n* = 3 samples). **g** Plot of $C_E$: $C_A$ mean grey value ratios for as-prepared, mineralized or decalcified multi-protocellular prototissues. Decalcification restores the grey value ratio of the pristine prototissue. Data are presented as mean values ± s.d. (*n* = 10 images). **h** Left, fluorescence CLSM image of a decalcified prototissue with intact $C_A$ (red) and $C_E$ (green) protocells and matrix (M, black) domains. Right, corresponding BF image. Scale bar, 50 μm. (**i**) Fluorescence intensity/grey value line profiles (white line in (**h**)). Blue shaded area: matrix (M); green shaded area: $C_E$ protocell; red shaded area: $C_A$ protocell. **j** Cryo-SEM images of a decalcified prototissue recorded from the interior of $C_A$ (left) or $C_E$ (right), and in the extra-protocellular space (M). Networks of silica particles (no ACP) are observed in $C_A$ and $C_E$. The decalcified matrix remains intact. Scale bars from left to right; 1, 1 and 1 μm. Source data are provided as a Source Data file.

essentially the same (Fig. 6g–i), suggesting that negligible amounts of ACP remained in the decalcified samples. Cryo-SEM images of the decalcified samples indicated that the Alg-MA matrix and silica networks within the colloidosomes remained intact after esterase-mediated acidification of the prototissues (Fig. 6j). Only low levels of calcium were detected in the matrix by SEM-EDX elemental mapping (Supplementary Fig. 43), consistent with dissolution of ACP. In contrast, control experiments using calcified multi-protocellular prototissues prepared from a 1:1 binary population of ALP- or bovine serum albumin (BSA)-containing colloidosomes remained mineralized after addition of EA (Supplementary Fig. 44). Time-dependent changes in optical transmittance and Ca content accompanying the end of each of four cycles of ALP-mediated calcification/esterase-mediated decalcification indicated that the efficiency of mineralization/

demineralization was maintained for two cycles, after which the levels of Ca retained within the multi-protocellular prototissue disks was reduced due to diminishing ALP activity (Supplementary Fig. 45a, b). Consequently, although the external pH decreased significantly from ca. 7 to 6 over 24 h in the first calcification/decalcification cycle, subsequent mineralization/demineralization cycles produced a pH drop above 6.5 (Supplementary Fig. 45c), indicating a gradual loss in the capacity of the embedded colloidosome community to generate a proton flux within the multi-protocellular prototissues.

## Discussion
Inspired by the orchestrated self-transformation of soft living tissues into mineral-matrix composites such as bone, in this paper we demonstrate the design and chemical construction of a protocell/

matrix-integrated tissue-like material capable of endogenous calcification and decalcification. The methodology, which is straightforward and adaptable, involves the chemical integration of single or binary populations of enzymatically active MA-colloidosomes within a continuous Alg-MA hydrogel to produce a macroscopic three-dimensional connective tissue-like construct with homogeneous or gradient microstructures. The prototissues typically have a Young's modulus ($E$) of ca. 0.2 MPa comparable to the stiffness of the soft dermis tissue of skin[46]. Site-specific deposition of calcium phosphate (ACP/HAP) within the protocells or throughout the extra-protocellular matrix, or both, is achieved in the presence of CaGP by modulating the phosphate reaction-diffusion gradient arising from ALP-mediated activity within the colloidosomes. Specifically, while mineralization of the MA-colloidosome/Alg-MA network occurs in both the discontinuous and continuous phases of the prototissue, calcification solely within the protocell interior or matrix is regulated by using a PEGDM hydrogel capable of inhibiting ACP nucleation, or a combination of colloidosome-entrapped PEG and a PAA-doped Alg-MA hydrogel to promote ACP nucleation. In each case, the extent of mineralization is dependent on the colloidosome number density, matrix crosslinking density and level of ALP activity, with a maximum mineral content of ca. 60 wt.% equivalent to values typical of cancellous bone[41]. Calcified prototissues with typical mineral contents of ca. 25 wt.% (10 vol.%) showed viscoelastic behaviour and were stiffer than their non-mineralized counterparts with $E$ values in the range of 2.16-4.18 MPa, which is comparable to cartilage in stiffness ($E$ = 250 kPa to 3 MPa for articulated cartilage)[47]. These values are considerably less than those for cancellous (1.41-3.47 GPa)[49] or cortical bone (ca. 10-22 GPa)[50], and some bone-mimicking composites[32,51], indicating that the calcified prototissues are too mechanically weak to function as bone replacement materials but could have potential uses as bioactive agents in facilitating bone regeneration. In particular, the chemo-mechanically active calcified gradient prototissues could potentially establish graded fields of bioactivity across musculoskeletal/cartilage interfaces for example.

Taken together, our approach provides a step towards the self-regulation of cytomimetic calcification within uniformly distributed or gradient distributed tissue-like micro-composites. The work contributes to the advancement of chemical materials research through the application of bottom-up synthetic biology, and could provide opportunities in bioinspired tissue engineering, hydrogel applications and bone biomimetics. In principle, it should be possible to control the ACP particle size by moderating the supersaturation levels achieved within the prototissues. Additionally, the transformation of ACP to crystalline HAP should be amenable to increased size/shape/texture regulation by changing the ACP concentration, pH or temperature, as well as by incorporating crystal growth additives in the hydrogel matrix. Moreover, the methodology presented in this work provides the opportunity to tune the polymorphic structure and phase behaviour of a range of inorganic minerals by finely adjusting and utilizing the synergistic chemical effects arising from the interplay between the enzyme-active colloidosomes and their surrounding hydrogel matrix. For example, other types of prototissue-mediated calcification, such as calcium carbonate biomineralization could be implemented by replacing ALP with carbonic anhydrase. Such an approach could open novel routes to the in situ regulation of microbiologically induced mineral precipitation for applications in bone healing[52] and concrete reinforcement[53].

Looking ahead, natural biomineralization mechanisms involving non-collagenous protein/matrix interactions and hierarchical structuration could be incorporated into the material design to increase the complexity of the prototissues. Moreover, it should be feasible to introduce complex communities of functionally targeted colloidosomes into the protocell/matrix constructs to establish diffusive chemical and biochemical communication pathways between living bone cells and unmineralized/mineralized prototissues. Construction of these hybrid materials would enable the on-site coordination of artificial signal transduction processes at bone tissue interfaces, providing important local control functions for the activation and degradation of living cells and their surrounding extracellular matrix. While the cytomimetic approach shows promise for inspiring new ideas in tissue engineering, the procedure is currently based on synthetic environments and is not intended to directly model physiological biomineralization. Thus, future studies on the biocompatibility, cell adhesion properties, viability, and biodegradation of protocell/matrix-integrated materials will be required to assess the translational potential of the calcified prototissues.

## Methods

### Characterization methods

*Optical and fluorescence microscopy* imaging was performed on a Leica DMI3000 B manual inverted fluorescence microscope. Images were analysed using ImageJ software (version 1.54f). *Confocal laser scanning microscopy (CLSM)* imaging was acquired on a Leica SP8 AOBS confocal laser scanning microscope attached to a Leica DM I8 inverted epi-fluorescence microscope. The microscope was equipped with 50 mW 405 nm diode laser (405 nm for Dylight405), a 65 mW Ar Laser (458, 476, 488, 496 and 514 nm lines, 488 nm for FITC excitation) and a 20 mW DPSS yellow laser (561 nm for RITC). Detection bands were set at 415-465 nm (Dylight405), 498-548 nm (FITC) and 571-621 nm (RITC). All measurements were performed at room temperature. Images were analysed using ImageJ software (version 1.54f). Three-dimensional reconstructions from Z stack scans were performed using Icy Software (version 2.5.2.0). Three-dimensional reconstruction images were used to calculate the volume fraction of colloidosomes in the prototissues using ImageJ software (version 1.54 f). Origin 2023b (Academic) was used to plot graphs.

*Scanning electron microscopy (SEM)*: Freeze-dried colloidosomes were mounted on aluminium stub with carbon tape and coated with a layer of silver. Second electron and back scattered electron images were collected on JEOL JSM-IT300 microscope at accelerate voltage 15 kV and work distance 10 mm.

*Energy-dispersive X-ray spectroscopy (EDX)* analysis was performed at accelerate voltage 15 kV and work distance 10 mm for elements calcium, oxygen, phosphorus, carbon and silicon. An intra-protocellular calcified prototissue disk was air-dried and a fractured surface generated by using a microtome, followed by coating with a layer of silver. Other samples, including homogenous calcified prototissues, matrix-calcified prototissue disks, gradient calcified prototissues and multi-protocellular prototissues were frozen in liquid ethane and then manually fractured in a cryo-SEM sample holder in liquid nitrogen followed by freeze-drying. The samples were coated with a layer of carbon before EDX analysis.

*Cryo-SEM:* Cryo-SEM imaging was performed using SEM Quanta 200 attached with a Gatan Alto 2500 cryo-SEM attachment that allows samples to be plunge-frozen in liquid nitrogen slush, freeze fractured, coated, and imaged at low temperature and high vacuum. A prototissue disk prepared in a 4.5 mm diameter x 1.6 mm depth sample holder was quickly frozen in liquid nitrogen and transferred into a pre-cooled sample chamber and freeze-fractured to generate a flat cross-section surface. 5 min sublimation was applied to generate a thin layer of the sample, followed by spray coating with silver before imaging. The imaging was performed at an acceleration of 12 kV and 10 mm working distance. Both back-scattered electron and second-electron images were recorded. Calcium phosphate particle sizes derived from cryo-SEM images were determined using ImageJ.

*Raman spectra* (in the region from 660 to 1220 cm$^{-1}$, centred at 950 cm$^{-1}$) were recorded using a Renishaw 2000 Laser Raman spectrometer (785 nm excitation) at room temperature. Raman spectra in the range of 915 to 985 cm$^{-1}$ were deconvoluted to one or two bands

corresponding to different phosphate vibration model and based on that the proportion of each calcium phosphate phase in a sample was calculated and plotted.

*Powder X-ray diffraction (PXRD)* patterns were recorded using a Bruker D8 Advance with a PSD LynxEye detector and Cu radiation (1.54 Å). A step size of 0.01 degree and 2 s per step in the 2-theta range from 10 to 70 degrees was applied for the measurement.

*Thermogravimetric analysis* was performed on a NETZSCH STA equipment (STA 449 F3 Jupiter) in a temperature range from 25 to 800 °C at 5 °C/min in air atmosphere with an air flow at 50 mL/min. The weight loss below 180 °C was attributed to the evaporation of water from the samples. The weight loss between 180 and 800 °C was associated with pyrolysis of the organic matrix. The remaining weight percentage at 800 °C was the inorganic residue. The organic/silica ratio ($R$) was acquired from the thermogravimetric curves of an as-synthesized (reference) prototissue. $R$ was calculated by the weight percentage (w%) change between 180 °C and 800 °C divided by the residual weight percentage:

$$R = \Delta w\%(180\ to\ 800\ °C)/w\%(800\ °C) \tag{1}$$

The inorganic component remaining at 800 °C in the calcified prototissue consisted of silica and calcium phosphate. The organic/inorganic ratio was first determined from the thermogravimetric curves, and the organic/silica reference (control) ratio was then used to calculate the organic, silica and calcium phosphate contents in the corresponding calcified prototissue samples:

$$w\%(CaP) = w\%(800\ °C) - w\%(silica) = w\%(800\ °C) - w\%(organic)/R \tag{2}$$

The organic/silica ratio in the reference sample and calcified sample was considered the same.

*Density characterizations* of the disk-shaped prototissue were performed using a Gas pycnometer (Micromeritics accuPyc 1340) fitted with a 1 cm³ sample chamber. The density, sample mass and volume were used to calculate the volume fraction of the calcified prototissues.

*Mechanical testing.* Unconfined compression testing was performed with samples of non-mineralized or calcified prototissue disks (diameter 4.5 mm x height 2/1.8 mm) using a uniaxial mechanical tester (Shimadzu) equipped with a 500 N load cell. A ramp compression at a speed of 0.3 mm/min was applied to record the stress-strain curves. Derivative of the stress-strain curve was used to determine the linear region (the constant region corresponding to the linear region of the stress-strain curve). The compressive modulus (Young's modulus) was calculated within the linear range. Three or four measurements were performed and standard deviations calculated.

*Rheological measurements.* Rheology measurements on samples (diameter 1.9 cm x height 1 mm) were performed with a NETZSCH kinexus pro+ rheometer equipped with PU20 geometry. The sample thickness was measured using a micrometre screw gauge and the thickness used to set up the gap for the strain amplitude sweep. This ensured that a comparable normal force applied on each prototissue sample tested. The elastic modulus (G′) and viscose modulus (G″) were determined at 0.1% complex shear strain in the linear viscoelastic region.

## Preparation of colloidosomes

20 mg of partially hydrophobic silica nanoparticles were dispersed in 4 mL anhydrous dodecane and sonicated for 5 min to produce a well-dispersed suspension. 200 μL of an aqueous solution containing enzymes (2 mg ml⁻¹ ALP in 0.1 M Tris buffer, pH 8.5; 25 wt.% FITC-labelled ALP or RITC-labelled ALP was used with ALP for samples prepared for CLSM imaging) were added to the oil dispersion followed by vigorous hand shaking for 1 min to obtain a water in oil Pickering emulsion. 30 μL of TMOS and 20 μL of TPM were added into the

emulsion and the emulsion was rotated for 48 h at room temperature. The resulting crosslinked methacrylate (MA)-functionalized colloidosomes were transferred into water by washing twice with 70% (v/v) ethanol/water, 50% (v/v) ethanol/water solution and Milli-Q-purified water using centrifugation at 555 × g for 2 min, and then concentrated by centrifugation at 555 × g for 2 min. Dispersions of MA-colloidosomes were kept at 4 °C for further use.

## Construction of protocell/matrix-integrated prototissue

A 4 %w/v Alg-MA solution (100 μL) was mixed with equal amounts of a concentrated ALP-containing MA-colloidosome dispersion and a small amount of 4 %w/v I2959 solution (photo initiator, dissolved at 70 °C, final I2959 concentration, 0.5 %w/v). In some experiments, 5 μL, 10 μL and 30 μL of 40 %w/v PEGDM were added to the mixture such that the weight percentage of PEG in the matrix corresponded to 25, 50 and 75 wt.%, respectively. The mixture was centrifuged at 555 x g for 2 min and the clear supernatant discarded. A small amount of the sedimented mixture (20 μL) was transferred to a home-made mould with a silicone isolator attached on one side of a microslide, covered by a coverslip on the other side, and then exposed to UV illumination (290-390 nm light; MAX-303 Xenon Light Source equipped with UV module) for 7 min to produce a covalently crosslinked protocell/matrix-integrated prototissue. The distance between the light emitter and the sample was set at 1.5 cm. The covalent-crosslinked construct was subsequently immersed in 0.1 M CaCl₂ solution for 1 h to implement Ca²⁺ crosslinking, and then washed three times with water to remove residual free Ca²⁺. Prototissues were kept at 4 °C for further use. Moulds were prepared for the fabrication of prototissue disks (4.5 mm diameter x 1.6/1.8/2.0 mm thick), rectangular strips (12 ×2 x 2 mm) or thin films (4.5 mm diameter x 260 μm).

## Endogenous calcification of prototissues

Prototissue disks (volume = 20 μL) fabricated with MA-colloidosomes containing ALP (2 mg ml⁻¹, 25 wt.% FITC-labelled ALP) were immersed in 8 mL of 0.05 M CaGP solution (in 0.2 M DEA buffer, pH = 9.8) and shaken at 10 rpm in an orbital shaker (Stuart SI50 Orbital Incubator, 25 ± 0.5 °C). The volume ratio between the prototissue and CaGP solution was 1 : 400. After 24 h, the mineralized samples were washed twice with DEA buffer and once with water. The calcified prototissues were cut into thin slices and then stained using 10 mg ml⁻¹ ARS aqueous solution for 20 min and gently washed with water for CLSM imaging.

## Determination of calcium phosphate content of calcified prototissues

An *o*-CPC based colorimetric method was used to determine the time-dependent increase in calcium phosphate content in the mineralizing prototissues. A series of prototissue disks were immersed in CaGP solution to initiate mineralization and samples removed after 0.5, 3, 5, 12, 24 and 36 h, and washed twice with 10 mL of DEA buffer and once with water. To extract calcium, 0.2 mL of 2.5 M HCl was added to the disks and left overnight at room temperature, followed by addition of 0.8 mL water. The equilibrium mixture was centrifuged at 9391 × g for 20 min and the supernatant (0.5 mL) collected for analysis. The calcium reagent was prepared at pH 11.5 from 2-amino-2-methylpropanol (AMP, 500 mM), orthocresolphthalein complexone (o-CPC, 32 μM), and 8-hydroxychinoline (0.275 mM). Calcium calibration curves ($y = -0.07472 + 24.8944x$ ($R^2 = 0.99$) were made using 5 mM CaCl₂ stock solution with the final concentration ranging from 0 to 0.062 mM. 400 μL of the calcium reagent was added to 5 μL of the sample solution. The mixture was vortex-mixed for 30 s and the absorbance at 573 nm measured by UV-Vis spectroscopy in a quartz cuvette (Further dilution of the sample solution was carried out as required to fit with the linear range of the calibration curve). All tests were conducted on three parallel samples ($n = 3$ samples) and standard error bars were calculated

## Calcium phosphate phase transformations in calcified prototissues

PXRD and Raman spectroscopy analysis was performed on calcified prototissue disks (i) after immersion in CaGP solutions over 24 h and up to 28 d, and (ii) after immersed in CaGP for 24 h followed by removal, two gentle washes with water and immersion in the same volume of pure water for up to 28 d. In each case, samples were removed at defined time intervals, carefully washed with water and then freeze-dried prior to PXRD and Raman spectroscopy studies.

## Site-specific prototissue mineralization

*Intra-protocellular calcification*. Equal volumes of a concentrated suspension of ALP-containing MA-colloidosomes (ALP, 2 mg ml$^{-1}$) and an aqueous solution of PEDGM (40 %w/v) were mixed and added with 4 % w/v I2959 photo initiator aqueous solution (dissolved at 70 °C, final I2959 concentration = 0.5 %w/v). The mixture was concentrated by centrifugation at 555 x $g$ for 2 min followed by exposure to UV light in a mould to afford a covalently integrated colloidosome/PEGDM prototissue disk. Protocell-localized ALP-mediated calcification in the presence of added CaGP was undertaken as described above. As-prepared mineralized samples were stained by ARS and imaged under CLSM.

*Extra-protocellular calcification*. 4 %w/v Alg-MA solution was mixed with 20 %w/v PAA solution to give a Alg-MA: PAA mass ratio of 30 : 70. The mixed solution was then added to an equal volume of ALP/ PEG-containing MA-colloidosomes (2 mg ml$^{-1}$ ALP; 40 mg ml$^{-1}$ PEG) and with a final I2959 photo initiator concentration of 0.5 %w/v. The colloidosome suspension was concentrated by centrifugation at 555 × $g$ for 2 min and the sedimented phase transferred to a mould and then photo-crosslinked by UV illumination for 7 min. The resulting prototissue disk was calcified in the presence of CaGP as described above.

## Preparation and calcification of gradient prototissues

A gradient prototissue was fabricated as follows. A suspension of ALP/ PEG (2 mg ml$^{-1}$ ALP and 40 mg ml$^{-1}$ PEG)-containing MA-colloidosomes was mixed with an equal volume of matrix components (Alg-MA/PAA, 4 %w/v Alg-MA with 20 %w/v PAA solution, final 70 wt.% PAA) and photo-initiator (final I2959 concentration = 0.5 %w/v) and poured into a home-made rectangular holder (12 x 2 x 2 mm) lying on its side and the suspension allowed to sediment under gravity for different time periods. The samples were then illuminated with UV light for 7 min to induce covalent matrix-crosslinking and protocell-matrix tethering to afford self-standing gradient prototissues.

For chemo-mechanical/calcification studies, samples prepared after 5 min sedimentation were placed in a plastic petri and immersed in 20 mL of 0.05 M GaGP solution in Tris buffer (0.2 M, pH=7.5). Movies of the deformation process were constructed from photographs that were recorded within the first 120 min at time intervals of 1 min using a digital camera (Canon EOS 500D). The samples in CaGP solution were then placed in an orbital shaker (25 ± 0.5 °C, 10 rpm) and left to calcify for 24 h. Non-calcified gradient prototissues exhibiting chemo-mechanical deformation were prepared by immersion in 20 mL of 0.05 M CaCl$_2$ in Tris buffer (0.2 M, pH=7.5) and processed as above. To investigate the chemical stability of the deformed prototissues, both the calcified and non-calcified gradient prototissues were transferred into 35 mL of 0.05 M EDTA (pH=7.4) and the changes in curvature with time determined using a digital camera. The curvature was approximated as being equal to *1/R*, where *R* was the radius of a circle fitted to the deformed prototissue.

## Endogenous decalcification of calcified multi-protocellular prototissues

A suspension of ALP (2 mg ml$^{-1}$, 25 wt.% RITC-labelled ALP)-containing MA-colloidosomes was mixed with a population of esterase (40 mg ml$^{-1}$, 25 wt.% FITC-labelled esterase)-containing MA-colloidosomes at ratio of 1 : 1 and the binary population integrated into a Alg-MA/PEGDM hydrogel (PEG-50wt.%) by photo-assisted covalent matrix-crosslinking and protocell-matrix tethering. 8 mL of 0.05 M CaGP solution (DEA buffer, 0.2 M, pH=9.8) was externally added to the prototissue disk (20 µL) to initiate ALP-mediated calcification for 12 h. The mineralized prototissue was washed with DEA buffer and water, and then transferred into 1 mL of 0.5 M ethyl acetate solution (5 mM Tris buffer, pH=7.5) to trigger esterase-driven proton-mediated decalcification. Multi-protocellular prototissues prepared form a binary population of colloidosomes loaded with ALP (2 mg ml$^{-1}$, 25 wt.% RITC-labelled ALP) or BSA (20 mg ml$^{-1}$, 25 wt.% FITC-labelled BSA) were employed as control samples. Time-dependent changes in appearance of the prototissues before and after calcification and decalcification at different time were captured by using a digital camera.

## Cycling of calcification/decalcification in multi-protocellular prototissues

A series of multi-protocellular prototissue-disks consisting of ALP-containing MA-colloidosome and esterase-containing MA-colloidosomes at a number ratio of 1 : 1 were prepared as described above. 8 mL of 0.05 M CaGP solution (0.2 M Tris buffer, pH = 8.5) was externally added to the prototissue disk (20 µL) to initiate ALP-mediated calcification for 12 h. The samples were then washed twice with Tris buffer and once with water followed by addition of 1 mL 0.5 M ethyl acetate (5 mM Tris buffer, pH = 7.5) to initiate decalcification over 24 h. After decalcification, the samples were washed with Tris buffer and water and then re-immersed into 8 mL of 0.05 M CaGP solution to initiate a new cycle of calcification followed by decalcification using the procedures described above. At the end of each calcification/decalcification cycle, samples were recorded using a digital camera (Canon EOS 500D).

The corresponding levels of mineral content at different stages in the cycle was determined as follows. 200 µL of 2.5 M HCl solution was added to each sample to extract calcium, followed by the addition of water to make up the volume to 1 mL. The suspension was then centrifuged at 9391 × $g$ for 20 min and the top 500 µL of supernatant isolated to determine the calcium concentration by using an o-CPC based colorimetric method (see above). The changes in pH during the first decalcification process, as well as the pH at the end of each decalcification process, were monitored.

## Statistics and reproducibility

Unless otherwise noted, each experiment reported in Fig. 1b, e (left), f; 2b-e,g; 3b-d,f,g; 4b,d,e,g; 5e; 6b,c,e,h,j was repeated at least three times independently with similar results.

## Reporting summary

Further information on research design is available in the Nature Portfolio Reporting Summary linked to this article.

## Data availability

All experimental data supporting the findings are available within the paper and in the Supplementary Information files. Source data is available for Figs. 1, 2, 3, 4, 5 and 6 and Supplementary Fig. 1, 2, 3, 4, 6, 7, 8, 9, 10, 11, 12, 13, 14, 15, 16, 17, 18, 19, 21, 22, 23, 24, 25, 26, 27, 28, 29, 30, 31, 32, 33, 34, 35, 36, 37, 38, 39, 40, 41, 42, 43, 44 and 45 in the associated source data file. Source data are provided with this paper.

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

## Acknowledgements
The authors thank the Wolfson Bioimaging Facility, Chemical Imaging Facility, Chemistry X-ray Facility and Bristol Composites Institute at the University of Bristol for help with physical characterizations. The authors thank Judith Mantell for help with cryo-SEM, Dr James Smith for Raman measurements, Dr Yusuf Mahadik for mechanical tests, Dr Ning Gao for help with protocell construction and alginate modification, and Drs Rafael Moreno Tortolero and Jean-Charles Eloi for useful discussions on mechanical testing and electron microscopy characterizations, respectively. We thank the Swedish Research Council for an international postdoctoral fellowship grant (2020-00747, R.S.), a Marie Sklodowska-Curie grant (101061428, R.S.) under Horizon Europe and the ERC Advanced Grant Scheme (EC-2016-674ADG 740235, S.M.) for financial support.

## Author contributions
R.S., M.L. and S.M. conceived the experiments. R.S. performed the experiments. Z.Y. assisted with thermogravimetric analysis calculations and calcification-decalcification cycling experiments. R.S., M.L. and S.M. undertook the data analysis. R.S., M.L. and S.M. wrote the manuscript. All authors, including M.M.S. contributed to project discussions and commented on the manuscript.

## Competing interests
The authors declare no competing interests.
