## [Transparent Peer Review file · Nature Communications]

Cytomimetic calcification in chemically self-regulated prototissues

Corresponding Author: Professor Stephen Mann

Version 0:

Reviewer comments:

Reviewer #1

(Remarks to the Author)

Mann et al report the design of a biomimetic system which allow a regulated biomineralization. There is a significant body of work performed in this paper, including elegant experiment, which are wealthy to be published. However, there is two major problems in this paper (see below) which makes it unsuitable for publication in Nat Comm in its current form, and requires, thus, profound modifications.

General comments

1/ The aim of the paper is unclear. There is no guiding thread to help the reader to easily follow (i) the purpose of the experiments, (ii) the strategy employed, and (iii) the methodology used to conduct these experiments. Overall, the reader encounters a significant number of experiments presented in succession, but without a clear logic. Sometimes, the aim of certain experiments is completely unclear, making the paper feel like a mere juxtaposition of experiments without a coherent "narrative". This is particularly surprising given the large number of experiments conducted.

2/ There is an obvious overstatement in the terms attributed to the objects studied in this paper. This includes: prototissues, protocells, colloidosomes, multi-protocellular prototissues, micro-composites ... The excessive use of these terms makes the article difficult to follow, as some of them are not appropriately suited to the objects designed/studied in the paper. The authors need to justify the use of these terms with solid arguments. More broadly, while I understand that this approach might be intended to give the study greater impact, in my opinion, it detracts from the article rather than enhancing it. Of course, this problem contributes to the one mentioned in point 1/.

Specific comments

The authors describe a biomimetic system based on the regulation of calcium phosphate mineralization regulated by enzymes (ALP). However, they did not cite any recent work in which ALP was used to perform as enzyme-assisted mineralization. I recommend to add, at least, the most recent references.

In the whole manuscript, the sentences are often too long and makes the paper difficult to follow.

When comparing the modulus in the text (l. 150, 187, 189, 219, 224, 227), the uncertainties of data have to be added. In fig 2a, 3a, 4a, 5a and 6a, scheme legends can be added to facilitate comprehension.

56-61: extracellular vesicle implication in inorganic phosphate generation and calcium phosphate deposition should be mentioned.

112: why an incubation at RT promotes ALP activity? This is not true. The authors have to justify. More importantly, the authors did not measure the enzymatic activity in this paper. This is a serious problem.

122: The conclusion about CaP is overinterpreted from these data. But, observations of 200nm particles could be linked with the literature regarding the nature of CaP particles?

165: indication (PEG) should be added.

189: MA-colloidosome/PEGDM-integrated prototissue at 35d should be compared to MA-colloidosome/Alg-MA prototissue at 35d and not at 28d.

199-203: Overinterpretation, phosphate production / enzymatic activity has never been measured.

258 : here again phosphatase activity is mentioned but never measured.

275: "and CaGP added" -> "and CaGP was added" ?

298: "possibly due to leakage", can TEM measurement confirm it?

320: here again, ALP activity is mentioned but never measured.

623 (fig 2h): legend saying that the amount of CaP is measured but this is the amount of Ca only!

668 (fig 5): 5a. in the legend, (i), (ii) are indicated but not shown in the graph. 5d. The x axis should correspond to the bars.

715 (fig 6d): why is there a red fluorescence intensity detected in the CE area ?

Supp material:

129 (fig S1): what does the third image represent?

133 (fig S2): c. Label too small, x interval should be larger, n should be mentioned like in other size distribution graphs.

142 / 120-124 (fig S3): labelling is performed before the permeabilization test but the fluorochrome could alter the normal diffusion of internal ALP. In what external solution is the test realized?

175 (fig S7): we just see the surrounding P and Ca of the buffer and Csome containing Si. How could they conclude these elements are inside Csome? T0 experiments needed?

187 (fig S8): n should be indicated; b is for the C region and c for the M region? Clarify.

344 (S25c): y axis scale should be the same in both graphs.

274 (fig S18): Same remark as for fig 2h, legend indicating that the amount of CaP is measured but this is the amount of Ca only!

330 (fig S24b): circles are added but totally biased the image reading.

415 (fig S31): why did the authors study the 10min sedimentation, why is it not shown in fig 5?

474 (fig S38): the object may bias the phosphor map if P is presents a lower intensity.

504 (fig S41): Why is the control 20mg/mL of BSA knowing that it is to compare it to 40mg/mL esterase ?

516 (fig S42): in the Si cartography, why does the Csome deform? Clarify

Reviewer #2

(Remarks to the Author)

The authors present a platform for biomineral synthesis. Silica colloidosomes suspended in a hydrogel matrix serve as artificial osteoblasts, capable of enzyme-driven calcium phosphate synthesis. The synthesis can be localized to both the dispersed and continuous phases, or exclusively one phase. using different polymers in the hydrogel which inhibit the mineralization. The authors also demonstrate how the mineralization can direct the physical properties of the hydrogel, such as the shape. The authors also present a mechanism to dissolve the minerals enzymatically, after mineralization has occurred. The distinct populations of calcifying and decalcifying colloidosomes are presented as artificial osteoblasts and osteoclasts, respectively. The process is fully reversible, though the authors note the gradual drop in pH could affect the overall lifetime.

The provided methodology is thorough and one could reasonably reproduce the authors' results. A variety of analytical techniques, including SEM, LSCM, and XRD, are presented to support the proposed structures and materials. The characterization and analysis is extensive, and the claims made by the authors are tenable. The text is clear, concise, and does an excellent job of explaining their findings.

How is this paper compares to other artificial biomineralization literature is unclear. The authors present their platform as a self-sustainable prototissue as opposed to an artificial cell or vesicle. Broadly speaking, though, the paper depicts a biphasic system capable of localizing calcium phosphate synthesis. The extent of the control over the synthesis, compared to contemporaries, is questionable. For instance, the paper does not make it clear whether the system can dictate the size, shape, and structure of the calcium phosphate particles. The authors note the structural differences between materials formed inside and outside the colloidosomes, but due not present a theory as to why. The efficacy of the system for biomineral synthesis is weakened by the apparent lack of control. Demonstration of tunability of the mineral, not just the hydrogel matrix, would strengthen the paper.

During the discussion, the authors propose using calcified hydrogel for bone regeneration applications. Their analysis of the material makes the possibility seem feasible, though the advantages of their approach are unclear; comparable materials could be produced with less complex systems. Advantages over pre-existing scaffold material would need to be demonstrated. In addition, limiting the mineral synthesis to calcium phosphate also limits their potential. The synthesis mechanics presented are not limited to hydroxyapatite, and colloidosomes could be modified with enzymes to produce different minerals. This would strengthen the paper by opening the door to a variety of applications aside from tissue engineering.

Overall, the paper does an excellent job detailing a system that is not appreciably distinct from its contemporaries. Improved tunability or diversity would enhance the overall impact.

Reviewer #3

(Remarks to the Author)

The work presented by Sun and coworkers is a valuable contribution to biomimetic materials. The study demonstrates innovative strategies for integrating biomineralisation processes into protocell/matrix systems, offering an interesting way to design biomimetic materials with regulable mechanical properties. However, several areas require further discussion and expansion to strengthen the manuscript. Below, I outline specific points for consideration:

1. The manuscript will benefit from discussing its relevance in the broader context of biomineralisation and material stiffening techniques. Comparisons with recent advances, such as (but not limited to) ion-doped nano-hydroxyapatite for chitosan scaffolds (10.1186/s13036-024-00458-9), biomineralised collagen composites (10.1016/j.msec.2019.110572), and hierarchical biomineral structures (10.1073/pnas.2120177119), should be included. Similarly, the authors must engage with studies on nanoscale control of mineralisation (10.1038/s41467-023-43733-x), stiffness enhancement in biomaterials (10.1021/acs.macromol.9b00124, 10.1039/D3NR05828J), polymer–ceramic systems (10.1021/acs.omega.0c00846), and bone-mimicking composites (10.1021/acs.biomac.3c01143), among others. Moreover, the work should be compared to microbiologically induced calcite precipitation, a widely used technique in both biological (bone healing, 10.1016/j.msec.2019.110572) and structural (concrete reinforcement, 10.1016/j.cemconres.2018.08.014) applications. Finally, recent developments in matrix-assisted protocell assembly (10.1002/adfm.202405781) (this work is particularly relevant due to its focus on organisation and rheological properties of synthetic tissues in a pseudo-ECM) and controlling environments and cell-free expression of biomineralising enzymes within polymeric artificial cells (10.1038/s41557-023-01391-y) require that the authors critically evaluate the novelty and impact.

The authors are invited to address the following points experimentally or, if they deem this unfeasible, rework and rephrase the claims of their paper to ensure adherence to the presented evidence.

2. The use of fluorescence imaging to quantify protein encapsulation is acceptable but suboptimal. Incorporating alternative protein quantification methods (e.g. BCA assay) would increase confidence
3. The SEM-EDX elemental mapping data, lacks quantitative analysis of key parameters such as the Ca/P ratio. Please incorporate and compare this under different conditions to strengthen the conclusions
4. TGA is used to estimate the (in)organic content but the methods for correcting baseline values, such as accounting for silica contributions, are not fully described, clarification is required
5. The reliance on Tris at pH 7.5 for biomineralisation lacks physiological relevance. Biological systems involve complex ionic environments, inorganic buffers like bicarbonate or phosphate, and regulatory proteins such as fetuin-A. The use of biologically relevant buffers, serum-supplemented media (DMEM or similar), and natural extracellular matrices (e.g., collagen) would provide more meaningful results. Furthermore, dynamic conditions (e.g., flow systems) should be considered to replicate transport effects that are important in tissue engineering.
6. While calcification achieves cartilage-like stiffness, they remain inadequate for load-bearing applications such as bone repair (requiring ~ 20 GPa). The absence of anisotropy or dynamic mechanical property analyses (for instance creep or fatigue testing) further limits the materials for real-world applications
7. A significant limitation is the absence of tests such as interaction with cells or biomolecules, evaluating biocompatibility and biological integration. Studies on cell adhesion, viability, or enzymatic degradation will assess its actual translational potential
8. No real data on the yield or production volumes of the materials. The material output (e.g. mineral content per volume of precursor or total material produced per reaction) needs to be quantified to give a better perspective on scalability and practicality
9. The spatial calcification patterns attributed to diffusion of CaGP and enzymatic conversion are observed but not quantified and/or modeled. Reaction-diffusion dynamics lack computational or experimental validation, so little can be concluded on controllability and reproducibility of the patterns

furthermore, integrating the following aspects would increase the impact of the study, but they are not strictly necessary if the other revisions are addressed with sufficient robustness.

10. While the study demonstrates a novel approach to biomineralisation, it would definitely benefit from a deeper exploration of how its methodology compares to natural biomineralisation mechanisms (see also point 1.), such as those involving osteocalcin or matrix gla protein, and how the resulting materials perform relative to natural tissues and other state-of-the-art biomimetic systems. Additionally, incorporating concepts from hierarchical structures in nature (e.g., bone or nacre) could further contextualise the work and enhance its relevance.

This study proposes an interesting approach to biomimetic materials, with innovative strategies for integrating biomineralisation into protocell/matrix systems. However, addressing the outlined points will significantly strengthen the manuscript. A better critical review of the state of the art, and resolving these issues experimentally, or revising the claims where necessary, should satisfactorily address my concerns and improve the work's stand.

Version 2:

Reviewer comments:

Reviewer #1

(Remarks to the Author)

The authors have really clarified all the terms that were unclear, so the manuscript is much better now. I think they have responded well to the questions raised. Moreover, they have done what was asked, which must have required a lot of experiments and, therefore, significant work. With the corrections, the figures seem more understandable to me.

One response that bothers me is the point 20/. I asked the authors whether the fluorochrome labeling of ALP could be responsible for the apparent lack of ALP leakage. They don't really answer to this specific point, except by saying that it is a method often used. That's true, but it's not a reason not to question the method. It is clear for me that this protocol induces a bias in the interpretation of the results.

I'm still convinced that there are too many unnecessary additional figures (Supp Mater), and they muddle the main message of the paper.

Reviewer #2

(Remarks to the Author)

Rebuttals to the individual responses are provided below:

Rebuttal 1: The additional context improves the introduction.

Rebuttal 2: The motivation behind the methodology is explained in the original text. The benefits, however, of the cytomimetic approach, compared to other mineralization systems, are not obvious. The text presents a complicated synthesis method, but doesn't explore the necessity for the complexity. Control over the synthesis would demonstrate a clear advantage, which is why it was suggested.

Rebuttal 3: The highlighted text is appreciated. Further text on the inhibition of ACP nucleation, like that in page 6, would strengthen the discussion section.

Rebuttal 4: See Rebuttal 2.

Rebuttal 5: See Rebuttal 2.

Rebuttal 6: See Rebuttal 2.

The author's willingness to expand the scope of their research is appreciated, but the advantages of their system are not any more obvious in the new text.

Reviewer #3

(Remarks to the Author)

I would like to commend the authors for their extensive rework of the study. Their revisions have significantly strengthened the manuscript. The improved framing and additional discussions have enhanced the overall readability and coherence of the study.

My only remaining concerns are:

--Physiological relevance (point 5)

"These interesting suggestions are beyond the scope of the current work. The work presented focuses on a chemical/cytomimetic approach to achieve different modes of calcification in tissue-like materials. Our studies do not aim to investigate the use of the materials for bone generation, although such studies would be an interesting area of future work."

The focus on a cytomimetic chemical approach is appreciated, and it fully is understandable that additional experiments in biologically relevant media may not be feasible within the scope or timeline of the current study. However, several statements in the manuscript suggest potential biomedical applications, including:

31 "bioinspired tissue engineering, hydrogel technologies and bone biomimetics"

348 "potential uses as bioactive agents in facilitating bone regeneration"

365 "Such an approach could open novel routes to the in situ regulation of microbiologically induced mineral precipitation"

373 "would enable the on-site coordination of artificial signal transduction processes at bone tissue interfaces"

etc.

If the work is not intended to directly model physiological biomineralization, the manuscript would benefit from a clearer distinction between its cytomimetic scope and its potential relevance to biomedical fields. A small textual revision could clarify that while the system obviously aims to inspire future work in tissue engineering, it has not yet been tested in physiologically relevant conditions and is currently validated only in synthetic environments. It would be important in order to provide the correct expectations and framing.

--Biocompatibility (point 7)

Similarly, the authors have acknowledged that biocompatibility and cell studies are beyond the scope of their work, and they

have included a brief outlook in the discussion. This is a good addition, but since they discuss bioinspired tissue engineering and bone biomimetics, it would be better to explicitly state that no biocompatibility or cell-based studies have been conducted in this work. This small clarification would prevent any overstatement of the current level of validation.

Besides that, I think that the manuscript is in a very good state and very close to acceptance.

Response letter:

“Cytomimetic calcification in chemically self-regulated prototissues” (NCOMMS-24-63977-T).

We thank all the referees for their time, positive recommendations and relevant comments on various aspects of the work. We have carefully revised the manuscript considering all suggestions by the reviewers shown in blue font below. The detailed point-by-point responses to the reviewers’ suggestions and comments with relevant changes made to the manuscript are provided below. A yellow marked-up version of the revised manuscript and supplementary information is also provided.

We hope that the revised version of the manuscript can now meet the reviewers’ expectations and can be considered for publication.

REVIEWER COMMENTS

Reviewer #1 (Remarks to the Author):

Mann et al report the design of a biomimetic system which allow a regulated biomineralization. There is a significant body of work performed in this paper, including elegant experiment, which are wealthy to be published. However, there is two major problems in this paper (see below) which makes it unsuitable for publication in Nat Comm in its current form, and requires, thus, profound modifications.

General comments

1/ The aim of the paper is unclear. There is no guiding thread to help the reader to easily follow (i) the purpose of the experiments, (ii) the strategy employed, and (iii) the methodology used to conduct these experiments. Overall, the reader encounters a significant number of experiments presented in succession, but without a clear logic. Sometimes, the aim of certain experiments is completely unclear, making the paper feel like a mere juxtaposition of experiments without a coherent “narrative”. This is particularly surprising given the large number of experiments conducted.

Response:

We have thoroughly revised the manuscript according to the referees’ comments to add additional context and clarify the aims and motivation of the work.

2/ There is an obvious overstatement in the terms attributed to the objects studied in this paper. This includes: prototissues, protocells, colloidosomes, multi-protocellular prototissues, micro-composites ... The excessive use of these terms makes the article difficult to follow, as some of them are not appropriately suited to the objects designed/studied in the paper. The authors need to justify the use of these terms with solid arguments. More broadly, while I understand that this approach might be intended to give the study greater impact, in my opinion, it detracts from the article rather than enhancing it. Of course, this problem contributes to the one mentioned in point 1/.

Response:

We have tried to use terminology that is becoming well-recognized in this emerging research field. As the field is multi-disciplinary and therefore can be viewed from several different contexts ranging from synthetic biology to materials science, we have used a range of terms to help connect with readers from various cognate disciplines.

Specific comments

1. The authors describe a biomimetic system based on the regulation of calcium phosphate mineralization regulated by enzymes (ALP). However, they did not cite any recent work in which ALP was used to perform as enzyme-assisted mineralization. I recommend to add, at least, the most recent references.

Response:

Reference 36 in the manuscript is about using ALP to mediate calcium phosphate particle formation in a two-phase microenvironment (Chemistry of Materials 31.24 (2019): 10243-10255). In addition, according to the reviewer's suggestion, we have also added in the main text on page 3 references two very recent works (Nature Chemistry 2024, 16(4), 564-574; ACS Appl. Mater. Interfaces 2024, 16, 33005–33020) related to ALP enzyme-assisted mineralization:

"To mimic the interplay between osteoblasts and osteoclasts intrinsic to bone remodelling, a binary population of colloidosomes capable of implementing a cycle of endogenous ALP-mediated calcification^{39, 45}..."

2. In the whole manuscript, the sentences are often too long and makes the paper difficult to follow.

Response:

Please see the response to the general comments, point 1.

3. When comparing the modulus in the text (l. 150, 187, 189, 219, 224, 227), the uncertainties of data have to be added.

Response:

We have added the error bar of each data correspondingly. The new text reads as:

Page 5: "...prototissue (0.2 ± 0.04 MPa) was increased to 1.13 ± 0.09 MPa and 2.20 ± 0.33 MPa."

Page 6: "... the protocells with ACP (24 h) or HAP (35d in water) ($E = 3.81 \pm 0.52$ and 4.18 ± 0.32 MPa..."

Page 6: "... similar conditions ($E = 1.13 \pm 0.09$ (24 h) and 2.20 ± 0.33 MPa (35 d))..."

Page 7: "...colloidosome/PEGDM hydrogel matrix ($E = 1.13 \pm 0.09$ MPa, unmineralized; MA-colloidosome/Alg-MA matrix, $E = 0.2 \pm 0.04$ MPa MPa)..."

Page 7: "... with a concomitant increase in the Young's moduli from 1.84 ± 0.28 to 2.76 ± 0.25 MPa..."

Page 8: "...increased to values of 1.13 ± 0.09 , 3.81 ± 0.52 and 1.84 ± 0.28 MPa in as-prepared calcified prototissues..."

Page 8: "These values increased respectively to 2.20 ± 0.33 , 4.18 ± 0.32 and 2.76 ± 0.25 MPa after..."

4. In fig 2a, 3a, 4a, 5a and 6a, scheme legends can be added to facilitate comprehension.

Response:

We have added the legends in Fig. 1a. In the following schemes in Fig. 2a, 3a, 4a, 5a and 6a we have only added additional new symbol labels and did not repeat the symbol labels defined in Fig. 1a. This is to keep all schemes brief and clear.

5. 56-61: extracellular vesicle implication in inorganic phosphate generation and calcium phosphate deposition should be mentioned.

Response:

A new sentence with reference has been added in the Introduction part (page 2) which reads as:

“Osteoblasts secrete bone matrix and small extracellular vesicles that are enzymatically active in generating inorganic phosphate ions. The influx and accumulation of phosphate ions and Ca^{2+} through membrane transporters induce the nucleation and subsequent growth of calcium phosphate inside the extracellular vesicles⁴⁴. Confinement within the vesicles together with the presence of the extracellular matrix operate synergistically to produce a mineralized tissue comprising an integrated network of living cells, organic matrix and inorganic nanocrystals.”

6/ 112: why an incubation at RT promotes ALP activity? This is not true. The authors have to justify. More importantly, the authors did not measure the enzymatic activity in this paper. This is a serious problem.

Response:

We agree that room temperature does not specifically promote ALP activity. We have adjusted the sentence in the manuscript on page 4 and now it reads as:

“Prototissue disks were incubated in an aqueous solution of CaGP at room temperature (25 ± 0.5 °C) to provide a reservoir of free Ca^{2+} ions and initiate ALP-mediated hydrolysis of GP to generate phosphate ions within the tissue-like constructs (Supplementary Fig. 6).”

Furthermore, we have undertaken ALP activity tests according to the reviewer’s suggestion. A new Supplementary Figure as shown below has been added to the SI.

New Supplementary Fig. 6. Activity of (a) free ALP in different aqueous buffers; (b) ALP after encapsulation in free colloidosomes (Csome-ALP) compared with free ALP in 0.2 M DEA buffer pH 9.8; and (c) colloidosome-encapsulated ALP within a prototissue (prototissue-ALP) compared with free Csome-ALP (n=3). The results show that ALP has the highest activity in 0.2 M DEA buffer (pH 9.8), followed by 0.2 M tris buffer (pH 8.5), 0.2 M tris buffer (pH 7.5) and Milli-Q water. ALP activity inside free colloidosomes shows a decreased activity compared with free ALP at the same conditions as in (b), possibly due to limited diffusion of the product as the colloidosomes sediment at the bottom of the quartz cuvette after 2 min. ALP activity in the prototissue shows lower activity compared with the free Csome-ALP within an initial period of 20 min (c). This can be attributed to the diffusion limitation placed on substrate penetration into the prototissue. After 20 min, activity in the prototissue becomes higher than in the free colloidosomes.

The new experimental procedure has been added into the Supplementary Information on page 4. It reads as:

“Alkaline phosphatase (ALP) activity:

Activity of free ALP and ALP after encapsulation within colloidosomes: 10 μL ALP solution (5 U/mL) together with a varied amount of p-nitrophenylphosphate (pNPP, 1 mM/5 mM) solution in buffer were added with additional buffer solution to make the final volume up to 1 mL (ALP 0.05 U/mL). The final pNPP concentration was in the range from 2 to 4000 μM . Enzyme activity was measured using a Cary 300 UV-

Vis spectrophotometer to record the change in absorbance at 410 nm. 0.2 M DEA buffer (pH 9.8), 0.2 M Tris buffer (pH 8.5 and pH 7.5) and Milli-Q water were used individually as the buffer/solvent. Activity of ALP encapsulated in colloidosomes was tested using the method described above. 10 μ L ALP containing colloidosome suspension (5 U/mL) together with varied amount of pNPP (1 mM/5 mM) solution in buffer were added with addition of 0.2 M DEA buffer solution pH 9.8 to make up the final volume to 1 mL (ALP 0.05 U/mL). To avoid scattering from the colloidosome suspension, the solution was allowed to equilibrate for 2 min before recording the change in absorbance at 410 nm. **Activity of ALP in colloidosomes assembled within prototissues:** 20 μ L of a tissue disk was placed in 8 mL 1 mM pNPP solution in 0.2 M DEA buffer pH 9.8 (ALP 0.05 U/mL). The sample was gently rotated on a rotator. At certain times (0/3/7/10/15/20/30/50/70 min), 100 μ L solution was removed from the reaction system after centrifugation at 3000 rpm for 30 s. The collected solution was added with additional 400 μ L 0.2 M DEA buffer pH 9.8 to make up to 500 μ L solution for UV-Vis spectroscopic analysis at 410 nm. To compare the ALP activity in free colloidosomes, 20 μ L of a concentrated colloidosome suspension prepared by centrifugation at 2000 rpm for 2 min was added to 8 mL 1 mM pNPP solution in 0.2 M DEA buffer pH 9.8 (ALP 0.05 U/mL) and monitored at 410 nm using a Cary 300 UV-Vis spectrophotometer.”

7. 122: The conclusion about CaP is overinterpreted from these data. But, observations of 200 nm particles could be linked with the literature regarding the nature of CaP particles?

Response:

As the particle size range can be broad and difficult to measure with high accuracy, we have reduced the number of significant figures in the values of the mean size quoted in the revised manuscript on page 4. As the CaP particle size is strongly related to the synthesis conditions employed, we did not specifically compare our measurements with CaP sizes recorded in the literature.

8. 165: indication (PEG) should be added.

Response:

We have added the indication (PEG) in the manuscript. It reads as:

“... we replaced the Alg-MA hydrogel with a poly(ethylene glycol, PEG) matrix produced by ...”

9. 189: MA-colloidosome/PEGDM-integrated prototissue at 35d should be compared to MA-colloidosome/Alg-MA prototissue at 35d and not at 28d.

Response:

The comparison between MA-colloidosome/PEGDM-integrated prototissue (at 35 d in water) and MA-colloidosome/Alg-MA prototissue (at 28 d in water) was described in the original manuscript because these were the times associated in both cases with the results on the crystallization of ACP to HAP. We have now carried out additional experiments to test the Young’s modulus of MA-colloidosome/Alg-MA prototissue after immersion in water for 35 d. The data are shown below. The Young’s modulus of MA-colloidosome/Alg-MA prototissue is 2.20 ± 0.33 MPa. We have replaced the previous data for 28 d with the new data for 35 d.

The new text in manuscript reads as:

Page 5: “... for the non-mineralized prototissue (0.2 ± 0.04 MPa) was increased to 1.13 ± 0.09 MPa and 2.20 ± 0.33 MPa when mineralized with ACP (24 h, 24 wt.%,) or HAP (35 d, 24 wt.%,)...”

Page 6: “The stress-strain curves for the MA-colloidosome/PEGDM-integrated prototissues showed evidence for viscoelasticity along with increased levels of stiffness when mineralized specifically inside the protocells with ACP (24 h) or HAP (35d in water) ($E = 3.81 \pm 0.52$ and 4.18 ± 0.32 MPa, respectively) (Supplementary Fig. 23-24) compared with homogeneously calcified MA-colloidosome/Alg-MA prototissues prepared under similar conditions ($E = 1.13 \pm 0.09$ (24 h) and 2.20 ± 0.33 MPa (35 d)), respectively (see Supplementary Fig. 16).”

The corresponding Supplementary Figures have been modified as shown below.

Supplementary Fig. 15. (a) Unconfined compression stress-strain curves of pristine (gray dashed line), calcified (ACP) (black line, 35 d) and calcified (HAP) (blue line, 35 d) MA-colloidosome/Alg-MA prototissues. (b) Overlapping plots of stress-strain curve of calcified (ACP) prototissue (black line) and its corresponding derivative (red).

Supplementary Fig. 16. Mechanical properties of non-mineralized (pristine), calcified (ACP, 24 h) and calcified (HAP, 35 d) prototissues measured by unconfined universal compression. The encapsulated ALP concentration is 2 mg/mL. Error bars represent standard deviation ($n = 3$).

10. 199-203: Overinterpretation, phosphate production / enzymatic activity has never been measured.

Response:

We have addressed this issue. Please see response to Specific Comment 6.

11. 258 : here again phosphatase activity is mentioned but never measured.

Response:

See response to Specific Comment 6.

12. 275: “and CaGP added” -> “and CaGP was added” ?

Response:

The text has been changed and now reads:

"... and CaGP was added to induce ALP-mediated ..."

13. 298: "possibly due to leakage", can TEM measurement confirm it?

Response:

We have revised the text because as the reviewer implies we do not have direct evidence for leakage. For the calcification-decalcification experiment, the same prototissue disk is used; first it is immersed in a calcification solution for 12 hours and then the calcified prototissue disk is cleaned and immersed in ethyl acetate solution to carry out decalcification. The cycle is then repeated using fresh solutions. Therefore, the reason for the reduced amount of Ca^{2+} amount in the sample is more likely attributed to the decreased activity of ALP after several cycles. We have refined the statement in the manuscript on page 10. It reads as:

"... after which the levels of Ca retained within the multi-protocellular prototissue disks was reduced due to diminishing ALP activity ..."

14. 320: here again, ALP activity is mentioned but never measured.

Response:

See response to Specific comment 6.

15. 623 (fig 2h): legend saying that the amount of CaP is measured but this is the amount of Ca only!

Response:

We have adjusted the legend which read as:

"(h) Plot showing amount of Ca^{2+} determined from acid-extracted prototissues as a function of calcification time; the Ca^{2+} measurements are used as a proxy for the amount of calcium phosphate deposited in the prototissue."

16. 668 (fig 5): 5a. in the legend, (i), (ii) are indicated but not shown in the graph. 5d. The x axis should correspond to the bars.

Response:

(1). (i), (ii) indicate two steps. (i) Gravity induced Csome gradient and (ii) Covalent crosslinking, which are clearly shown with the arrow in the middle of scheme in Fig. 5a. (2). According to the reviewer suggestion, we have adjusted Fig. 5d as shown as below:

17. 715 (fig 6d): why is there a red fluorescence intensity detected in the CE area ?

Response:

We have re-analysed the CLSM images and also repeated the experiment. It was found that all the C_E population (green population) showed slight amounts of red fluorescence (typical intensity *ca.* 21 to 33) for the uncalcified sample, calcified sample and decalcified sample (see additional Figure A, below). This indicates that the weak red fluorescence detected in the C_E population is not due to leaking of ALP from the C_A population, which is also verified by the results in Supplementary Fig. 3 where the colloidosome shows high retention of ALP. The weak red fluorescence intensity detected in the C_E populations is due to the low levels of autofluorescence of the C_E population.

We have added a sentence to note the slight red fluorescence in C_E populations in the legend of Fig. 6d. It reads as:

“Background levels of red fluorescence are observed in C_E due to autofluorescence in the samples.”

Additional Figure A. CLSM images of uncalcified/calcified/decalcified prototissues. From left to right columns are red channel (RITC labelled ALP), green channel (FITC labelled esterase), merged fluorescence images, bright field image and plot of red/green fluorescence intensity along the white dashed line in the third column.

Supp material:

18. 129 (fig S1): what does the third image represent?

Response:

Supplementary Fig. 1 shows SEM images of lyophilized MA-modified colloidosomes. The first two images show the silicified outer membrane and the third image shows the internal silica network of a broken colloidosome.

We have updated the Figure and legend as follows:

“**Supplementary Fig. 1.** Scanning electron microscopy (SEM) images of lyophilized MA-modified colloidosomes showing silicified outer membrane (a, b) and internal silica network (c).”

19. 133 (fig S2): c. Label too small, x interval should be larger, n should be mentioned like in other size distribution graphs.

Response:

We thank reviewer for pointing out the issue. We have changed the size of label in Supplementary Fig. 2c and added the count number as well. The new Supplementary Fig. 2 is shown as below:

20. 142 / 120-124 (fig S3): labelling is performed before the permeabilization test but the fluorochrome could alter the normal diffusion of internal ALP. In what external solution is the test realized?

Response:

The dye labelling is used to distinguish the distribution of ALP. The experiment was carried out in aqueous solution. We mixed colloidosome aqueous suspension with either dye-labelled dextran (Supplementary Fig. 3a) with varied molecular weight or dye-labelled ALP aqueous solution (Supplementary Fig. 3c,d) to ascertain if the external molecules penetrate into the colloidosome and therefore to evaluate the membrane permeability. This is a widely used method.

Furthermore, FTIC-labelled ALP or RITC-labelled ALP was taken up by the colloidosomes and encapsulated within the colloidosome interior with negligible leaking. We attributed this to the hierarchical and adsorbent interior silica structure of the colloidosome (as shown in Supplementary Fig. 1), which has been documented in a previous publication (Nature Communications 12.1 (2021): 6113.).

21. 175 (fig S7): we just see the surrounding P and Ca of the buffer and Csome containing Si. How could they conclude these elements are inside Csome? TO experiments needed?

Response:

1). The calcified sample was fractured under liquid nitrogen followed by freeze drying. The cross-section is displayed in Supplementary Fig. 8a (original Fig. S7), which clearly shows the Csome (C) population embedded in a continuous matrix (M). The Csome is not surrounded by buffer.

2). The line profile in Supplementary Fig. 8b shows the Ca, P Si element distributions along the yellow dashed line that crosses the colloidosome (white dashed circle). Ca and P element intensities are comparable along the line, which means that the Ca and P levels are comparable between matrix and colloidosome interior. Si is detected in Csome (C). If there is no Ca, P inside the Csome, then the Csome (C, red dashed circles area) should have no Ca/P signal and appear black.

3). Supplementary Fig. 8 also corresponds to Fig. 2. In Fig. 2e, CaP particles in both the colloidosome (C) and matrix (M); thus, the results in Supplementary Fig. 8 and Fig. 2e are consistent with each other.

To make it clearer, we have revised Supplementary Fig. 8 (original Fig. S7) as shown as below:

22. 187 (fig S8): n should be indicated; b is for the C region and c for the M region? Clarify.

Response:

A revised Supplementary Fig. 9 (Original Fig S8) is shown below: we have added the counting number to Supplementary Fig. 9b,c correspondingly, and clarified Supplementary Fig. 9b and 9c according to the suggestion of the reviewer.

“Supplementary Fig. 9. Cryo-SEM monitoring of prototissue calcification. (a) Images recorded from the colloidosome interior (C), extra-proto-cellular Alg-MA matrix (M) and at the C/M interface 0, 0.5, 3 and 24 h after addition of CaGP. Calcium phosphate particles are mainly observed in C after 0.5 h and appear initially in the matrix adjacent to C, and then throughout M after 3 h. Dense aggregates of calcium phosphate are observed with different textures in both C and M at 24 h. (b,c) Plots of time-dependent increases in calcium phosphate particle size **in the C (b) and M regions (c) of the prototissue.**”

23. 344 (S25c): y axis scale should be the same in both graphs.

Response:

The figure has been updated according to the suggestion of reviewer as shown below:

“Supplementary Fig. 26. (original Fig S25).”

24. 274 (fig S18): Same remark as for fig 2h, legend indicating that the amount of CaP is measured but this is the amount of Ca only!

Response: We have revised the legend of Supplementary Fig. 19 (original Fig S18) which reads as below:

“Supplementary Fig. 19. Plot showing amount of Ca^{2+} determined from acid-extracted prototissues prepared with 0 wt.% and 50 wt.% PEG in a Alg-MA/PEGDM matrix; the Ca^{2+} measurements are used as a proxy for the amount of calcium phosphate deposited in the prototissue.”

25. 330 (fig S24b): circles are added but totally biased the image reading.

Response:

We have updated the figure as well as the legend. The new Supplementary Fig. 25 (Original Fig S24) is shown below:

“Supplementary Fig. 25. (a-c) Bright field (BF) images of calcified ALP-containing MA-colloidosome/Alg-MA/PAA prototissue films prepared with PAA proportions of 0 (a), 55 (b) and 71 wt.% (c) in the extra-protecellular matrix (corresponding total [COOH] = 0.23 M (Alg only), 0.55 M (Alg + 55 wt.% PAA) and 0.94 M (Alg + 71 wt.% PAA). In each case, films were mineralized for 24 h in the presence of CaGP. Calcification occurs both within the colloidosomes and matrix but with higher relative levels of colloidosome calcification (decreased light transmittance) in the absence of PAA (a). Mineralization occurs at a similar level in both colloidosomes and surrounding matrix (similar transmittance values) at PAA = 55 wt.% (b); preferential calcification of the matrix occurs at PAA = 71 wt.% (c). (d) Colloidosome: matrix gray value ratios for calcified prototissues constructed with different amounts of PAA in the Alg-MA matrix. Error bars represent standard deviation (n = 5 to 8). Gray values derived from BF images were determined using ImageJ software. Lower gray values correspond to decreased light transmittance and increased calcification. No intra-protocellular PEG was used in the experiments. Error bars represent standard deviation (n = 5-7).”

26. 425. 15 (fig S31): why did the authors study the 10 min sedimentation, why is it not shown in fig 5?

Response:

Fig. 5b,c,d show the formation of a gradient prototissue with a distinctive gradient at low sedimentation time that gradually separates into a well-delineated matrix-enriched layer and Csome-enriched layer structure at around 7 min. The microstructure of samples prepared at longer sedimentation times of 10 min is the same as that observed at 7 min. We used the sample prepared after 10 min sedimentation as an example to demonstrate the microstructure difference with the sample prepared after 5 min sedimentation, with Cryo-SEM image shown in Supplementary Fig. 32 (Original Fig S31) and SEM-EDX data shown in Supplementary Fig. 33.

In addition, we have added the fluorescence image of the sample prepared after 10 min of sedimentation to Fig. 5b. The corresponding layer ratio of the sample has been included in Fig. 5d. The revised Fig. 5 is shown below:

27. 474 (fig S38): the object may bias the phosphor map if P is presents a lower intensity.

Response:

1). We have conducted SEM-EDX tests for all gradient prototissues using the same method and same conditions. Therefore, the SEM-EDX results of different samples can be compared in parallel to determine the difference between each other.

2) As for sample in Supplementary Fig. S39 (original Fig S38), the lower level of P is reasonable. Firstly, it is a CaCl₂ immersed sample. The Ca²⁺ ions are locked into the matrix by chelation with the carboxyl groups of the Alg-MA/PAA matrix (confirmed by Supplementary Fig. 37 and Supplementary Fig. 40). Additionally, no phosphate ions are generated in the system. Therefore, high Ca²⁺ and low/neglected P signal are detected in this sample (Supplementary Fig. 39).

28. 504 (fig S41): Why is the control 20mg/mL of BSA knowing that it is to compare it to 40mg/mL esterase ?

Response:

The prototissue-2A40E is capable of decalcification because esterase (E) transforms ethyl acetate to acetic acid which dissolves CaP mineral in the samples. For the control experiments, BSA is used together with ALP because BSA has no esterase-like activity and does not dissolve the formed CaP mineral. Thus, it is not strictly necessary to have identical BSA and esterase concentrations given the inactivity of the control samples.

According to the reviewer comments, we have carried out additional experiments to compare the ability of calcification-decalcification of three prototissue, prototissue-2A20B, 2A40B and 2A40E. The photograph below (Additional Figures B and C) shows how the three prototissue samples appear as-prepared, after calcification and after decalcification. All the three samples are capable of calcification, and transform from translucent to opaque after calcification. However, only prototissue-2A40E becomes translucent after decalcification. The other two, prototissue-2A20B and 2A40B remained opaque, which means there is not signification mineral dissolution in these two samples. We also monitored the pH after decalcification and determined the Ca^{2+} amount in the samples after each process. The data is shown below. The results indicated that prototissue-2A20B and prototissue-2A40B do not have decalcification capability.

Additional Figure B. Photograph of as-prepared/calcified/decalcified prototissue-2A20B, prototissue-2A40B and prototissue-2A40E. The three translucent prototissues became opaque after calcification. The yellow colouration of prototissue-2A40E is because of the entrapped esterase enzyme. After decalcification process, only prototissue-2A40E turns translucent, which indicates the mineral is dissolved. The prototissue-2A20B and prototissue 2A40B samples remained relatively opaque.

Additional Figure C. (a) Plot of pH recorded after decalcification in three prototissue systems showing only a decrease in pH to *ca.* 5.5 in prototissue-2A40E (esterase-containing). (b) Plot of amount of Ca^{2+} determined from acid-extracted samples (Ca measurement as a proxy calcium phosphate deposited in a prototissue) showing that the Ca^{2+} amount is comparable (*ca.* 10 μM) in the three prototissues and increases to *ca.* 40 μM after calcification. Decreases in Ca^{2+} were only detected in prototissue-2A40E after decalcification.

An additional sentence has been added to the legend of Supplementary Fig. 42 (Original Fig S41) to clarify the use of the BSA control. It reads as:

“Supplementary Fig. 42. ...Additional results showed that the BSA concentration had no effect on the decalcification results.”

29. 516 (fig S42): in the Si cartography, why does the Csome deform? Clarify

Response:

The sample is a decalcified sample and was frozen in liquid ethane and then manually fractured in liquid nitrogen to produce a cross-section, followed by freeze drying. In the absence of CaP mineral, the difference in texture and microstructure between the organic matrix and Csome (inorganic silica) gives rise to differential shrinking during the drying process, which gives rise to deformation of the relatively soft and elastic Csome. This can also be seen in Fig. 4g.

Reviewer #2 (Remarks to the Author):

The authors present a platform for biomineral synthesis. Silica colloidosomes suspended in a hydrogel matrix serve as artificial osteoblasts, capable of enzyme-driven calcium phosphate synthesis. The synthesis can be localized to both the dispersed and continuous phases, or exclusively one phase. using different polymers in the hydrogel which inhibit the mineralization. The authors also demonstrate how the mineralization can direct the physical properties of the hydrogel, such as the shape. The authors also present a mechanism to dissolve the minerals enzymatically, after mineralization has occurred. The distinct populations of calcifying and decalcifying colloidosomes are presented as artificial osteoblasts and osteoclasts, respectively. The process is fully reversible, though the authors note the gradual drop in pH could affect the overall lifetime.

The provided methodology is thorough and one could reasonably reproduce the authors' results. A variety of analytical techniques, including SEM, LSCM, and XRD, are presented to support the proposed structures and materials. The characterization and analysis is extensive, and the claims made by the authors are tenable.

The text is clear, concise, and does an excellent job of explaining their findings.

1/ How is this paper compares to other artificial biomineralization literature is unclear. The authors present their platform as a self-sustainable prototissue as opposed to an artificial cell or vesicle. Broadly speaking, though, the paper depicts a biphasic system capable of localizing calcium phosphate synthesis.

Response:

We have extended the introduction section to include additional references and context regarding relevant literature on artificial biomineralization systems. Please see responses to Reviewer 3, comment 1.

2/ The extent of the control over the synthesis, compared to contemporaries, is questionable. For instance, the paper does not make it clear whether the system can dictate the size, shape, and structure of the calcium phosphate particles.

Response:

In this paper we present a “cytomimetic” approach in which artificial protocells are employed as rudimentary models of osteoblasts and osteoclasts to mediate spatially localized calcification/decalcification in a hydrogel matrix. Our focus is on the development of an operating system rather than the fine tuning of parameters such as calcium phosphate size, shape and structure, which can be addressed in future research. In general, spherical amorphous calcium phosphate (ACP) particles are formed for each of the different calcification scenarios presented due to the supersaturation conditions associated with the Ca^{2+} concentration used and ALP activities employed to generate phosphate ions in situ. In principle, it should be possible to control the ACP particle size by moderating the supersaturation levels achieved within the prototissues. Additionally, the transformation of ACP to crystalline HAP should be amenable to increased size/shape/texture regulation by changing the ACP concentration, pH or temperature, as well as by incorporating crystal growth additives in the hydrogel matrix. We have included these considerations in the Discussion section; the new text on age 11 reads:

“In principle, it should be possible to control the ACP particle size by moderating the supersaturation levels achieved within the prototissues. Additionally, the transformation of ACP to crystalline HAP should be amenable to increased size/shape/texture regulation by changing the ACP concentration, pH or temperature, as well as by incorporating crystal growth additives in the hydrogel matrix.”

3/ The authors note the structural differences between materials formed inside and outside the colloidosomes, but due not present a theory as to why.

Response:

We have proposed the following reasons in different parts of the manuscript to explain the experimental observations concerning structural differences (see green highlights below).

Page 4: “After 24 h, mineralization within the colloidosomes – that is, **in closer proximity to the site of ALP-mediated phosphate production** – resulted in calcified silica networks that were relatively dense and consisted of aggregated calcium phosphate particles, 197 nm in mean size (Fig. 2e,f). In contrast, **efflux of excess phosphate into the surrounding Alg-MA matrix** produced smaller calcium phosphate aggregates (mean size, 146 nm) that were closely associated with the continuous polysaccharide network (Fig. 2e,f).”

Page 6: “Deposition of calcium phosphate solely within the interior of the colloidosomes was accomplished indirectly **by inhibiting ACP nucleation** in the surrounding hydrogel matrix. To achieve this, we replaced the Alg-MA hydrogel with a poly(ethylene glycol, PEG) matrix produced by photo-assisted crosslinking of PEG dimethacrylate (PEGDM, $M_w=750$; monomer units, $n=13$).”

Page 7: “Specifically, a carboxylate-rich polymer (polyacrylic acid, PAA, 100 kDa) was included in the Alg-MA hydrogel **to promote calcification** within the matrix while PEG (100 kDa) was encapsulated within the ALP-containing MA-colloidosomes **to inhibit intra-protocell nucleation and growth of ACP (Fig. 4a)**. As for the specific calcification inside the colloidosomes, the CaP size is much larger, around 506 nm. We attributed this **to the inhibitive effect of PEG** based matrix on calcification.

Discussion, page 10: “Site-specific deposition of calcium phosphate (ACP/HAP) within the protocells or throughout the extra-protocellular matrix, or both, is achieved in the presence of CaGP **by modulating the phosphate reaction-diffusion gradient arising from ALP-mediated activity within the colloidosomes**. Specifically, while mineralization of the MA-colloidosome/Alg-MA network occurs in both the discontinuous and continuous phases of the prototissue, calcification solely within the matrix or protocell interior **is regulated by using a PEGDM hydrogel or a combination of a PAA-doped Alg-MA hydrogel and colloidosome-entrapped PEG, respectively**. In each case, the extent of mineralization is dependent on the **colloidosome number density, matrix crosslinking density and level of ALP activity**, with a maximum mineral content of ca. 60 wt.% equivalent to values typical of cancellous bone.⁴¹”

4/ The efficacy of the system for biomineral synthesis is weakened by the apparent lack of control. Demonstration of tunability of the mineral, not just the hydrogel matrix, would strengthen the paper.

Response:

Please see response 2 above (reviewer 2) concerning the focus and potential of our work. In addition, it should be feasible to achieve other types of prototissue-mediated calcification such as calcium carbonate mineralization by replacing ALP with carbonic anhydrase for example in future studies. We have added this point to the Discussion, which reads as:

Page 11: “**Moreover, the methodology presented in this work provides the opportunity to tune the polymorphic structure and phase behaviour of a range of inorganic minerals by finely adjusting and utilizing the synergistic chemical effects arising from the interplay between the enzyme-active colloidosomes and their surrounding hydrogel matrix. For example, other types of prototissue-mediated**

calcification such as calcium carbonate biomineralization could be implemented by replacing ALP with carbonic anhydrase.”

5/ During the discussion, the authors propose using calcified hydrogel for bone regeneration applications. Their analysis of the material makes the possibility seem feasible, though the advantages of their approach are unclear; comparable materials could be produced with less complex systems. Advantages over pre-existing scaffold material would need to be demonstrated. In addition, limiting the mineral synthesis to calcium phosphate also limits their potential. The synthesis mechanics presented are not limited to hydroxyapatite, and colloidosomes could be modified with enzymes to produce different minerals. This would strengthen the paper by opening the door to a variety of applications aside from tissue engineering.

Response:

Thank you for this interesting and useful comment. We have added additional sentences in the Discussion to highlight this point:

Page 11: *“Moreover, the methodology presented in this work provides the opportunity to tune the polymorphic structure and phase behaviour of a range of inorganic minerals by finely adjusting and utilizing the synergistic chemical effects arising from the interplay between the enzyme-active colloidosomes and their surrounding hydrogel matrix. For example, other types of prototissue-mediated calcification such as calcium carbonate biomineralization could be implemented by replacing ALP with carbonic anhydrase.”*

6/ Overall, the paper does an excellent job detailing a system that is not appreciably distinct from its contemporaries. Improved tunability or diversity would enhance the overall impact.

Response:

Please see our responses above.

Reviewer #3 (Remarks to the Author):

The work presented by Sun and coworkers is a valuable contribution to biomimetic materials. The study demonstrates innovative strategies for integrating biomineralisation processes into protocell/matrix systems, offering an interesting way to design biomimetic materials with regulable mechanical properties. However, several areas require further discussion and expansion to strengthen the manuscript. Below, I outline specific points for consideration:

1A. The manuscript will benefit from discussing its relevance in the broader context of biomineralisation and material stiffening techniques.

Response:

To broaden the context, we have added additional text and references to the revised manuscript as follows.

1B: Comparisons with recent advances, such as (but not limited to) ion-doped nano-hydroxyapatite for chitosan scaffolds (10.1186/s13036-024-00458-9), biomineralised collagen composites (10.1016/j.msec.2019.110572), and hierarchical biomineral structures (10.1073/pnas.2120177119), should be included. Similarly, the authors must engage with studies on nanoscale control of mineralisation (10.1038/s41467-023-43733-x),

Response:

On page 2: *“Recent advances in the fabrication of bone-like composites have involved ion-doped nano-hydroxyapatite and chitosan scaffolds³¹, biomineralized bioglass scaffolds³², hierarchical biomineral structures³³ and the nanoscale control of mineralisation³⁴. From a cytomimetic perspective...”*

1C: ...stiffness enhancement in biomaterials (10.1021/acs.macromol.9b00124, 10.1039/D3NR05828J), polymer–ceramic systems (10.1021/acsomega.0c00846), and bone-mimicking composites (10.1021/acs.biomac.3c01143), among others.

Response:

Page 10: *“These values are considerably less than those for cancellous (1.41-3.47 GPa)⁴⁹ or cortical bone (ca. 10-22 GPa)⁵⁰, and some bone-mimicking composites^{32, 51}, indicating that the calcified prototissues are too mechanically weak to function as bone replacement materials but could have potential uses as bioactive agents in facilitating bone regeneration.”*

1D: Moreover, the work should be compared to microbiologically induced calcite precipitation, a widely used technique in both biological (bone healing, 10.1016/j.msec.2019.110572) and structural (concrete reinforcement, 10.1016/j.cemconres.2018.08.014) applications.

Response:

Page 11. *“Such an approach could open novel routes to the in situ regulation of microbiologically induced mineral precipitation for applications in bone healing⁵² and concrete reinforcement⁵³.”*

1E: Finally, recent developments in matrix-assisted protocell assembly (10.1002/adfm.202405781) (this work is particularly relevant due to its focus on organisation and rheological properties of synthetic tissues in a pseudo-ECM) and controlling environments and cell-free expression of biomineralising enzymes within polymeric artificial cells (10.1038/s41557-023-01391-y) require that the authors critically evaluate the novelty and impact.

Response:

Page 2: *“Prototissues capable of mechanical and chemical communication have been developed by matrix-assisted assembly of protocells⁴¹.”*

Page 2: *“Cell-free expression of biomineralization enzymes has been undertaken to initiate biomineralization within polymeric artificial cells³⁹...”*

2. The authors are invited to address the following points experimentally or, if they deem this unfeasible, rework and rephrase the claims of their paper to ensure adherence to the presented evidence.

The use of fluorescence imaging to quantify protein encapsulation is acceptable but suboptimal. Incorporating alternative protein quantification methods (e.g. BCA assay) would increase confidence.

Response:

The use of fluorescence imaging to quantify protein encapsulation is widely used in many papers (Nature chemistry 10.11 (2018): 1154-1163; Nature Chemistry 16.2 (2024): 158-167; Advanced Materials (2024): 2404607); we therefore believe that this methodology is acceptable given the overall focus and context of our work. Moreover, the encapsulated protein concentration in the colloidosome cannot be easily determined using the BCA method because of extraction problems associated with strong irreversible protein-silica interactions.

3. The SEM-EDX elemental mapping data, lacks quantitative analysis of key parameters such as the Ca/P ratio. Please incorporate and compare this under different conditions to strengthen the conclusions

Response:

We thank the reviewer for their valuable suggestion. We have included the Ca/P ratios to the SEM-EDX data for the different calcification scenarios. The corresponding changes in text have been made in the revised manuscript:

Page 4: *“The Ca/P ratio (1.84) was higher than reported ratios for amorphous calcium phosphate (1.0-1.5) and hydroxyapatite (1.67), possibly because of excess Ca²⁺ ions trapped in the Alg-MA matrix.”*

Page 6: *“The Ca/P elemental ratio was 1.6, consistent with the lower levels of Ca²⁺ in the sample.”*

Page 7: *“SEM-EDX elemental mapping gave a Ca/P ratio of 2.0 and indicated...”*

Supplementary Information, Page 21 Supplementary Fig. 31: *“Decreasing levels of Ca and P are detected in the top area of the sample, which could be due to the diffusion limitation of Ca and P into the top layer. The Ca/P ratio (2.48) in the top area is higher than the ratio measured in the calcified bottom region (2.02), which is attributed to the elevated Ca²⁺ levels associated with the matrix.”*

Supplementary Information, Page 22 Supplementary Fig. 33: *“The calcified sample prepared after 10 min sedimentation showed similar characteristics to the calcified prototissue prepared after 5 min sedimentation (see Figure S30). The top area shows a higher Ca/P ratio (2.61), consistent with decreasing levels of P due to matrix-induced diffusion constraints into the top layer.”*

Supplementary Information, Page 25 Supplementary Fig. 39: *“Minimal levels of P were detected, indicating the absence of calcium phosphate mineralization. The detected Ca was associated with the binding of Ca²⁺ ions to carboxyl groups in the Alg-MAA/PAA matrix.”*

Furthermore, a table of SEM-EDX derived elemental compositions for different samples has been added to the Supplementary Information as Supplementary Table 2:

Supplementary Table 2: SEM-EDX analysis of different calcified prototissues.

	Ca	P	O	C	Si	Ca/P
Supplementary Fig. 8, Alg-MA/Csome	7.0	3.8	14.1	7.6	1.6	1.84
Fig. 3, PEGDM/Csome; PEG 100wt%	9.9	6.2	29.6	52.2	0.5	1.60
Fig. 4, Alg-MA/PAA/Csome	17.4	8.6	39	22.3	6.2	2.02
Supplementary Fig. 31., Gradient 5 min, Top	7.2	2.9	16.3	14.4	1.4	2.48
Supplementary Fig. 31., Gradient 5 min, Bottom	8.0	4.0	15.0	11.7	2.1	2.00
Supplementary Fig. 33., Gradient 10 min, Top	6.0	2.3	23	18.7	1.1	2.61
Supplementary Fig. 33., Gradient 10 min, Bottom	6.2	3.2	18.9	13.3	3.2	1.94
Supplementary Fig. 39., Gradient 5 min, ref Top	3.8	0.0	18.8	20.0	1.9	-
Supplementary Fig. 39., Gradient 5 min ref, Bottom	2.9	0.0	19.7	21	5.3	-

4. TGA is used to estimate the (in)organic content but the methods for correcting baseline values, such as accounting for silica contributions, are not fully described, clarification is required

Response:

We have added more details about the methods used in the Supplementary Information under the Characterization section. The new text reads:

"The organic/silica ratio (R) was acquired from the thermogravimetric curves of an as-synthesized (reference) prototissue. R was calculated by the weight percentage (w%) change between 180 °C and 800 °C divided by the residual weight percentage: $R = \Delta w\% (180 \text{ to } 800 \text{ °C}) / w\%(800 \text{ °C})$. The inorganic component remaining at 800 °C in the calcified prototissue consisted of silica and calcium phosphate. The organic/inorganic ratio was firstly determined from the thermogravimetric curves, and the organic/silica reference (control) ratio was then used to calculate the organic, silica and calcium phosphate contents in the corresponding calcified prototissue samples:

$w\%(CaP) = w\%(800\text{ }^{\circ}C) - w\%(silica) = w\%(800\text{ }^{\circ}C) - w\%(organic)/R.$

The organic/silica ratio in the reference sample and calcified sample was considered the same.

5. The reliance on Tris at pH 7.5 for biomineralisation lacks physiological relevance. Biological systems involve complex ionic environments, inorganic buffers like bicarbonate or phosphate, and regulatory proteins such as fetuin-A. The use of biologically relevant buffers, serum-supplemented media (DMEM or similar), and natural extracellular matrices (e.g., collagen) would provide more meaningful results. Furthermore, dynamic conditions (e.g., flow systems) should be considered to replicate transport effects that are important in tissue engineering.

Response:

These interesting suggestions are beyond the scope of the current work. The work presented focuses on a chemical/cytomimetic approach to achieve different modes of calcification in tissue-like materials. Our studies do not aim to investigate the use of the materials for bone generation, although such studies would be an interesting area of future work.

6. While calcification achieves cartilage-like stiffness, they remain inadequate for load-bearing applications such as bone repair (requiring ~ 20 GPa). The absence of anisotropy or dynamic mechanical property analyses (for instance creep or fatigue testing) further limits the materials for real-world applications

Response:

We acknowledge the lack of strength of our calcified prototissue and the uniformity of many of the materials produced in the Discussion section

Page 11: *“These values are considerably less than those for cancellous (1.41-3.47 GPa)⁴⁹ or cortical bone (ca. 10-22 GPa)⁵⁰, and some bone-mimicking composites^{32, 51}, indicating that the calcified prototissues are too mechanically weak to function as bone replacement materials but could have potential uses as bioactive agents in facilitating bone regeneration.”*

Page 11: *“Taken together, our approach provides a step towards the self-regulation of cytomimetic calcification within uniformly distributed or gradient distributed tissue-like micro-composites.”*

7. A significant limitation is the absence of tests such as interaction with cells or biomolecules, evaluating biocompatibility and biological integration. Studies on cell adhesion, viability, or enzymatic degradation will assess its actual translational potential

Response:

These important considerations are beyond the scope of the current work, which concerns the development of a chemical/cytomimetic approach to achieve different modes of calcification in tissue-like materials. Our studies are exploratory and do not aim to address the actual translational potential. We have included the following comment in the Discussion section:

“Future studies on the biocompatibility, cell adhesion properties, viability, and biodegradation of protocell/matrix-integrated materials will be required to assess the translational potential of the calcified prototissues.”

8. No real data on the yield or production volumes of the materials. The material output (e.g. mineral content per volume of precursor or total material produced per reaction) needs to be quantified to give a better perspective on scalability and practicality

Response:

These interesting questions of production, scale-up and practicality are beyond the scope of the current work. However, in response to the reviewer's comments, we calculated the productivity of a mineralized prototissue as the mineral content per volume of the prototissue precursor. For the prototissue prepared from 2 mg/mL ALP enzyme, the CaP mineral yield per volume of the precursor is $0.04953 \pm 0.005 \text{ g/cm}^3$. It should be noted that the mineral content is also related to the enzyme concentration (Supplementary Table 1).

The corresponding information has been added in the manuscript on page 5.

Page 5: *"The calcium phosphate yield per volume of the prototissue was estimated as $0.04953 \pm 0.005 \text{ g/cm}^3$."*

9. The spatial calcification patterns attributed to diffusion of CaGP and enzymatic conversion are observed but not quantified and/or modeled. Reaction-diffusion dynamics lack computational or experimental validation, so little can be concluded on controllability and reproducibility of the patterns

Response:

The spatial calcification patterns are controlled at the experiment level with highly reproducibility. The reaction-diffusion dynamics can be observed from the change of fine microstructure during calcification process (Supplementary Fig. 9), but as the Reviewer points out detailed computational/experimental validations will be required to give greater control over material the processing. This is beyond the scope of the current work.

10. furthermore, integrating the following aspects would increase the impact of the study, but they are not strictly necessary if the other revisions are addressed with sufficient robustness.

While the study demonstrates a novel approach to biomineralisation, it would definitely benefit from a deeper exploration of how its methodology compares to natural biomineralisation mechanisms (see also point 1.), such as those involving osteocalcin or matrix gla protein, and how the resulting materials perform relative to natural tissues and other state-of-the-art biomimetic systems. Additionally, incorporating concepts from hierarchical structures in nature (e.g., bone or nacre) could further contextualise the work and enhance its relevance.

Response:

We thank reviewer for the comments. These are valuable points to take into consideration to design advanced tissue-like materials based on the methodology developed in this study. We have added new text in the Discussion section on page 11, which reads as:

"Looking ahead, natural biomineralization mechanisms involving non-collagenous protein/matrix interactions and hierarchical structuration could be incorporated into the material design to increase the complexity of the prototissues."

Response letter:

“Cytomimetic calcification in chemically self-regulated prototissues” (NCOMMS-24-63977-B).

We sincerely thank all the reviewers for their time and positive recommendations on our first revision. We have carefully re-revised the manuscript, incorporating all additional suggestions and comments which are highlighted in blue font below. A detailed point-by-point response to each comment, along with the corresponding revisions made to the manuscript, is provided below. A green-marked version of the revised manuscript is also included.

We hope that this further revised version meets the reviewers’ expectations and is now suitable for publication.

REVIEWER COMMENTS

Reviewer #1 (Remarks to the Author):

The authors have really clarified all the terms that were unclear, so the manuscript is much better now. I think they have responded well to the questions raised. Moreover, they have done what was asked, which must have required a lot of experiments and, therefore, significant work. With the corrections, the figures seem more understandable to me.

1. One response that bothers me is the point 20/. I asked the authors whether the fluorochrome labelling of ALP could be responsible for the apparent lack of ALP leakage. They don’t really answer to this specific point, except by saying that it is a method often used. That’s true, but it’s not a reason not to question the method. It is clear for me that this protocol induces a bias in the interpretation of the results.

Response:

(i) The high retention of ALP within the colloidosome (i.e., the apparent absence of ALP leakage) is attributed to the dense internal silica network of the colloidosome, as shown in Supplementary Figure 1. The effective retention of a range of enzymes due to residual silica nanoparticles within colloidosomes has been previously reported in *Nature Communications* (12.1, 2021: 6113). This point is covered in the main text on page 3, which reads:

*“...Although the MA-modified membrane was permeable to free ALP in the external environment, leakage of encapsulated ALP from the colloidosomes was minimal (**Supplementary Fig. 3**) due to irreversible binding of the enzyme to the silica membrane and internal silica network.”*

(ii) The data in Supplementary Figure 3 highlight that the labelled ATP readily crosses the membrane from the external environment (Supplementary Figure S3c,d) but does not leach out even though the label is still intact (Supplementary Figure S3e,f), consistent with irreversible binding of the dye-labelled ATP to silica inside the colloidosomes.

(iii) We tested the above hypothesis by undertaking new experiments with RITC-, FITC- or Dylight 650-labelled BSAs; the results showed that in each case BSA freely permeated into the colloidosome but once inside showed high levels of retention (see below).

Additional Figure: Fluorescence images showing uptake and retention of RITC-labelled BSA (a), FITC-labelled BSA (b) and Dy650-labelled BSA (c) into colloidosomes.

(iv) We also used a bicinchoninic acid (BCA) assay to determine the amounts of encapsulated unlabelled ALP or FITC-labelled ALP released from the same quantity of colloidosomes when placed in an identical volume of Milli-Q water (250 µL) for 24 hours. In both cases, the percentages of protein released were very small and almost identical (2.29 and 2.72 µg/mL for ALP and FITC-labelled ALP, respectively), confirming that the presence of the fluorophore had negligible effect on the leakage properties.

Response Figure: (a) Calibration curve made from BSA standard solution using the BCA assay. (b) Plot of released ALP from colloidosomes containing ALP or colloidosomes containing FITC-labelled ALP showing similar release data.

2. I'm still convinced that there are too many unnecessary additional figures (Supp Mater), and they muddle the main message of the paper.

Response:

We have kept all the SI figures as we consider that providing this information is important for a detailed assessment and appreciation of our work by the scientific community.

Reviewer #2 (Remarks to the Author):

Rebuttal 1: The additional context improves the introduction.

Rebuttal 2: The motivation behind the methodology is explained in the original text. The benefits, however, of the cytomimetic approach, compared to other mineralization systems, are not obvious. The text presents a complicated synthesis method, but doesn't explore the necessity for the complexity. Control over the synthesis would demonstrate a clear advantage, which is why it was suggested.

Response:

We previously provided an extensive response and revised the discussion to try and respond constructively to this comment, including where elements of control could be achieved. Whether a cytomimetic approach is superior to more conventional procedures is difficult to answer as the former remains a proof-of-concept while the latter is well developed; much more work is required to develop the former before a comparable evaluation can be made.

Our previous revision included the following statements:

"In principle, it should be possible to control the ACP particle size by moderating the supersaturation levels achieved within the prototissues. Additionally, the transformation of ACP to crystalline HAP should be amenable to increased size/shape/texture regulation by changing the ACP concentration, pH or temperature, as well as by incorporating crystal growth additives in the hydrogel matrix."

"Site-specific deposition of calcium phosphate (ACP/HAP) within the protocells or throughout the extra-protocellular matrix, or both, is achieved in the presence of CaGP by modulating the phosphate reaction-diffusion gradient arising from ALP-mediated activity within the colloidosomes. Specifically, while mineralization of the MA-colloidosome/Alg-MA network occurs in both the discontinuous and continuous phases of the prototissue, calcification solely within the protocell interior or matrix is regulated by using a PEGDM hydrogel with effect of inhibition of ACP nucleation or a combination of a PAA-doped Alg-MA hydrogel capable of accelerating ACP nucleation and colloidosome-entrapped PEG, respectively. In each case, the extent of mineralization is dependent on the colloidosome number density, matrix crosslinking density and level of ALP activity, with a maximum mineral content of ca. 60 wt.% equivalent to values typical of cancellous bone.⁴¹"

Rebuttal 3: The highlighted text is appreciated. Further text on the inhibition of ACP nucleation, like that in page 6, would strengthen the discussion section.

Response:

We thank reviewer for the suggestion. We have added additional text related to the inhibition of ACP nucleation in the Discussion section which reads as below:

Discussion, page 10: *"Specifically, while mineralization of the MA-colloidosome/Alg-MA network occurs in both the discontinuous and continuous phases of the prototissue, calcification solely within the protocell interior or matrix is regulated by using a PEGDM hydrogel capable of inhibiting ACP nucleation, or a combination of colloidosome-entrapped PEG and a PAA-doped Alg-MA hydrogel to promote ACP nucleation."*

Rebuttals 4,5,6:

Response:

Please see the response to Rebuttal 2.

Reviewer #3 (Remarks to the Author):

I would like to commend the authors for their extensive rework of the study. Their revisions have significantly strengthened the manuscript. The improved framing and additional discussions have enhanced the overall readability and coherence of the study.

My only remaining concerns are:

--Physiological relevance (point 5)

"These interesting suggestions are beyond the scope of the current work. The work presented focuses on a chemical/cytomimetic approach to achieve different modes of calcification in tissue-like materials. Our studies do not aim to investigate the use of the materials for bone generation, although such studies would be an interesting area of future work."

The focus on a cytomimetic chemical approach is appreciated, and it fully is understandable that additional experiments in biologically relevant media may not be feasible within the scope or timeline of the current study. However, several statements in the manuscript suggest potential biomedical applications, including:

31 "bioinspired tissue engineering, hydrogel technologies and bone biomimetics"

348 "potential uses as bioactive agents in facilitating bone regeneration"

365 "Such an approach could open novel routes to the in situ regulation of microbiologically induced mineral precipitation"

373 "would enable the on-site coordination of artificial signal transduction processes at bone tissue interfaces"

etc.

If the work is not intended to directly model physiological biomineralization, the manuscript would benefit from a clearer distinction between its cytomimetic scope and its potential relevance to biomedical fields. A small textual revision could clarify that while the system obviously aims to inspire future work in tissue engineering, it has not yet been tested in physiologically relevant conditions and is currently validated only in synthetic environments. It would be important in order to provide the correct expectations and framing.

Response:

We thank reviewer for the suggestion. We have refined the text in Abstract and Discussion to clarify that physiologically relevant conditions is not tested in this work.

Abstract: Page 31: "...and **could provide** potential opportunities in bioinspired tissue engineering, hydrogel technologies and bone biomimetics"

Discussion: Page 11: "**While the cytomimetic approach shows promise for inspiring new ideas in tissue engineering, the procedure is currently based on synthetic environments and is not intended to directly model physiological biomineralization. Thus, future studies on the biocompatibility, cell adhesion**

properties, viability, and biodegradation of protocell/matrix-integrated materials will be required to assess the translational potential of the calcified prototissues.”

--Biocompatibility (point 7)

Similarly, the authors have acknowledged that biocompatibility and cell studies are beyond the scope of their work, and they have included a brief outlook in the discussion. This is a good addition, but since they discuss bioinspired tissue engineering and bone biomimetics, it would be better to explicitly state that no biocompatibility or cell-based studies have been conducted in this work. This small clarification would prevent any overstatement of the current level of validation.

Response:

We thank reviewer for the suggestion; please see response and new text as shown above.